## Registered report

ecology/environmental science

primary production, coral reef fish, carbon, nitrogen, stable isotope analysis, pelagic energetic subsidies

**Author for correspondence:**
Ronan C. Roche
e-mail: r.roche@bangor.ac.uk

# Linking variation in planktonic primary production to coral reef fish growth and condition

Ronan C. Roche[1], Adel Heenan[1], Brett M. Taylor[2], Jill N. Schwarz[3], Michael D. Fox[4,5], Lucy K. Southworth[1,6], Gareth J. Williams[1] and John R. Turner[1]

[1]School of Ocean Sciences, Bangor University, Menai Bridge, Anglesey LL59 5AB, UK
[2]Marine Lab, University of Guam, Mangilao 96923, Guam
[3]School of Biological and Marine Sciences, University of Plymouth, Plymouth PL4 8AA, UK
[4]Woods Hole Oceanographic Institution, Woods Hole, MA 02543, USA
[5]Red Sea Research Center, King Abdullah University of Science and Technology, Thuwal 23955, Saudi Arabia
[6]Centre of Excellence for Coral Reef Studies, College of Science and Engineering, James Cook University, Douglas, QLD 4811, Australia

RCR, 0000-0002-6342-9571; JNS, 0000-0002-3589-3887; GJW, 0000-0001-7837-1619

Within low-nutrient tropical oceans, islands and atolls with higher primary production support higher fish biomass and reef organism abundance. External energy subsidies can be delivered onto reefs via a range of physical mechanisms. However, the influence of spatial variation in primary production on reef fish growth and condition is largely unknown. It is not yet clear how energy subsidies interact with reef depth and slope. Here we test the hypothesis that with increased proximity to deep-water oceanic nutrient sources, or at sites with shallower reef slopes, parameters of fish growth and condition will be higher. Contrary to expectations, we found no association between fish growth rate and sites with higher mean chlorophyll-a values. There were no differences in fish $\delta^{15}N$ or $\delta^{13}C$ values between depths. The relationship between fish condition and primary production was influenced by depth, driven by increased fish condition at shallow depths within a primary production 'hotspot' site. Carbon $\delta^{13}C$ was depleted with increasing primary production, and interacted with reef slope. Our results indicate that variable primary production did not influence growth rates in planktivorous *Chromis fieldi* within 10–17.5 m depth, but show site-specific variation in reef physical characteristics influencing fish carbon isotopic composition.

# 1. Introduction

Coral reefs are among the most productive global ecosystems, supporting highly abundant reef organisms and fish biomass, despite occurring within oligotrophic tropical oceans. This productivity paradox has been attributed to diverse processes, such as the uptake and recycling of nutrients between the coral host and their endosymbiotic algae [1], the efficient recycling of dissolved inorganic matter by reef sponges [2] and the presence of nitrogen-fixing cyanobacteria within reef corals [3]. However, the ecological importance and variety of mechanisms that deliver external nutrient subsidies stimulating benthic and planktonic food chains on coral reefs are increasingly recognized [4–7].

The variation in pelagic primary productivity across global coral reefs is strongly linked to the spatial and temporal variability in processes delivering external nutrient subsidies. Terrestrial connectivity is a source of external nutrients, via riverine sources creating broad gradients in nutrient availability [8], or derived from guano produced by resident seabird populations on isolated oceanic atolls [9]. Hydrodynamic connectivity and nutrient input from oceanic water masses can occur when physical mechanisms transport the higher nutrient concentrations present in cooler deep water layers upslope to shallower reefs [10,11].

A variety of related and potentially interacting physical mechanisms, such as upwelling [12], internal waves [11,13] and internal tidal bores [10,14] can transport these deeper waters onto coral reefs. These processes are further influenced by geophysical reef characteristics. For example, reefs with a gradual sloping bathymetry show a greater nearshore enhancement in phytoplankton biomass in atolls and islands across the Pacific [5]. This is thought to be explained by physical processes such as internal waves propagating more easily across shallower reef slopes, but being reflected from steeper slopes [5,15]. Once these cold nutrient-rich waters have reached shallower reef areas, breaking surface waves, wind-driven flow and tides can drive further transport into spur and groove systems [16] or across the reef crest [13]. Geographic location, season and the interactions between transportation mechanisms and variable reef slope and topography [17] all determine which areas of reef are influenced by deep-water nutrients, across scales ranging from metres to kilometres [18–20].

There is interest in how external nutrient inputs on coral reefs could relate to resistance and recovery from coral bleaching events [21,22]. One mechanism is through the ability of mixotrophic corals to favour heterotrophic feeding in response to increased resource availability at greater depth and proximity to allochthonous nutrient sources [23,24]. This coral trophic flexibility has been explained across scales by satellite-derived chlorophyll-a estimates, which correlate strongly with primary production throughout the photic zone [5,21]. Whether coral reef fishes exhibit similar flexibility in their feeding strategy in response to variability in primary production is unclear, although Hanson et al. [6] showed selective feeding on oceanic zooplankton by planktivorous reef fish. Terrestrial nutrient subsidies delivered by seabirds onto shallow reefs can positively influence reef fish biomass and growth rate [9]. It is likely that the spatial variation in nutrient delivery from deep-water oceanic sources will have similarly important and widespread effects on coral reef fish populations, but empirical evidence of this link is scarce.

Stable isotope ratios in fish tissue can be used to infer reliance on the multiple potential nutrient sources (e.g. terrestrial, deep-water and benthic) driving trophic pathways on a reef [7,25,26] and have been successfully employed to identify feeding zones [27], to detect habitat-level [28], and depth-related differences within food chains [29,30]. In particular, enriched $\delta^{15}N$ and depleted carbon isotope $\delta^{13}C$ ratios occur in the tissues of some reef fish with increasing concentrations of oceanic primary production [31,32].

Here we test the hypothesis that with increased depth and proximity to deep-water oceanic allochthonous nutrient sources, fish growth and condition will increase, and this pattern will be further emphasized in areas naturally higher in primary production. Specifically, we predict that planktivorous fish growth rate and condition (tissue lipid content assessed by C : N ratio; [33]) will be positively associated with high mean chlorophyll-a values and shallower slopes at reef sites. Additionally, we predict that planktivorous fish collected at greater depths will show enriched $\delta^{15}N$ and depleted $\delta^{13}C$ values indicating an increased reliance on primary production derived from deeper oceanic nutrient sources.

# 2. Methods

## 2.1. Sampling design

We sampled three atolls within the Chagos Archipelago, central Indian Ocean spanning approximately 170 km of latitude, from the 10 to 26 April 2019. Previous research within the Archipelago found

significantly different planktivorous fish biomass between atolls [34]. Within each atoll, we haphazardly selected six sites located on the seaward reef slopes of the atolls so that sites captured the principal cardinal directions of the atoll's coastline and were separated by at least approximately 1 km. We avoided locating sampling sites at the entrance passes of each atoll where exchange of water between the lagoon and ocean would probably complicate the physical processes driving productivity patterns [24].

## 2.2. Data and materials collected

To characterize the physical steepness of the reef slope, a transect of depth recordings was collected at each sampling site. The transect was performed with a hand-held depth transponder at each study site perpendicular to the reef slope, starting at approximately 5 m depth, and taking sequential, georeferenced depth readings, (Garmin Montana 650T ±3 m accuracy). Paired (GPS and depth transponder) readings were taken at approximately 30 s intervals moving offshore until the maximum depth range of the transponder was exceeded (approx. 70 m).

To collect particulate organic matter (POM) and zooplankton samples to determine a characteristic isotopic signature at each site we used the following procedure: seawater (approx. 10 l) was collected from 8 to 10 m depth, while zooplankton were collected from approximately 1 m depth using a net with a mesh size of 200 µm. The net was towed at idle speed (25 hp engine) for 10–15 min per site covering approximately 100–300 m. Both sample types were filtered through pre-combusted (450°C, 5 h) glass fibre filters (GF/F 0.7 µm, Whatman), which were then frozen at –20°C and transported to Woods Hole Oceanographic Institution, USA for storage.

To obtain data on fish growth rates and isotopic signatures, we collected otoliths and tissue samples from a ubiquitous reef-associated planktivorous Indian Ocean species *Chromis fieldi* which feeds on zooplankton, generally within a metre of the reef [35]. Individuals were collected by SCUBA divers using 5% clove oil in ethanol solution and hand nets at two depths categorized as moderate (approx. 17.5 m) and shallow (approx. 10 m). We aimed to collect a maximum of 10 individuals per depth at each site. Fish were euthanized in clove oil solution and kept on ice prior to processing within chilled Icey-Tek© cool boxes. The total length and fork length of each fish was recorded. Sagittal otoliths from each fish were extracted and stored for later analysis. Tissue samples from one site were damaged during ship movements at sea. The anterior dorsal muscle of each fish was sampled and dried at 60°C, before being frozen and transported to Bangor University, UK for storage.

## 2.3. Analyses

### 2.3.1. Stable isotopes

POM and zooplankton samples were thawed, rinsed with dilute HCl (5%) to remove carbonate material and rinsed in deionized water. Samples were dried and 2 mg of material were weighed into tin capsules for analysis. Three replicate samples for $\delta^{13}C$ and $\delta^{15}N$ were prepared for POM and zooplankton. Dried fish muscle tissue was ground into a powder and 0.5–1.5 mg of tissue weighed into tin capsules. Isotope $\delta^{13}C$ and $\delta^{15}N$ analysis of the collected fish tissue took place at the University of New Mexico Center for Stable Isotopes. Analysis was carried out using a Thermo Scientific Delta V mass spectrometer with a dual inlet and Conflo IV interface connected to a Costech 4010 elemental analyser and a high-temperature conversion elemental analyser.

Isotope values of $\delta^{13}C$ and $\delta^{15}N$ were reported using delta notation, in per mil (‰), as deviations from standards (Vienna Pee Dee Belemnite (V-PDB) for $\delta^{13}C$ and atmospheric $N_2$ for $\delta^{15}N$) according to the formula

$$\delta X_{\text{sample}} = \left[ \left( \frac{R_{\text{sample}}}{R_{\text{standard}}} \right) - 1 \right] \times 1000,$$

where $R_{\text{sample}}$ is the ratio of heavy to light isotope in the sample and $R_{\text{standard}}$ is the ratio of heavy to light isotope in the standard. Within-run analytical error assessed via repeated analysis of internal proteinaceous reference materials (Pugel and Acetanilide) was estimated to be ±0.2‰ for both $\delta^{13}C$ and $\delta^{15}N$.

The ratio of carbon to nitrogen per cent weight in tissue (C : N) from stable isotope analysis was compared among sites as a proxy for tissue lipid content [33]. Lipids are depleted in $\delta^{13}C$ relative to other tissue types and a positive linear relationship has been established between C : N and lipid content for aquatic animals [36]. If C : N values were above 3.5 [37], isotope $\delta^{13}C$ values from fish

muscle tissue were corrected for high lipid content using the formula from Post *et al*. [36]

$$\delta^{13}C_{normalized} = \delta^{13}C_{untreated} - 3.32 + 0.99 \times C:N.$$

In interpreting our results, two potential sources of variation in $\delta^{13}C$ and $\delta^{15}N$ and $C:N$ ratios were tested: (i) a potential influence of fish size on $C:N$ ratio was tested by a linear regression between $C:N$ ratio and FL using all fish samples and (ii) growth effects on $\delta^{13}C$ and $\delta^{15}N$ were tested by a linear regression between site level $K_{max}$ and site level $\delta^{13}C$ and $\delta^{15}N$ values (electronic supplementary material).

We used $\delta^{13}C$ and $\delta^{15}N$ values to characterize an isotopic niche as a proxy for the ecological niche occupied by *Chromis fieldi* across our sampling locations. Isotope niche was estimated by calculating the extent of $\delta^{13}C$ and $\delta^{15}N$ in biplot space using the area of standard ellipses estimated by Bayesian inference with the SIBER R package [38]. Standard ellipses (SEA$_c$) were fitted for fish, POM and zooplankton samples, grouping by atoll and depth, using a 40% confidence interval [38]. The size of ellipses was compared fitting Bayesian models (SEA$_b$: $10^4$ iterations), differences in standard ellipse area were interpreted graphically, and considered significant when greater than or equal to 95% of posterior draws for one group were smaller than the other [38].

### 2.3.2. Fish otolith analysis

Otolith microstructure was examined to determine fish growth rate following Taylor *et al*. [39]. One otolith from each pair obtained from individual fish was weighed to the nearest 0.00001 g and attached to a glass microscope slide using thermoplastic glue. Otolith sections were examined on a minimum of two occasions using a dissecting microscope with transmitted light. Ages in years were determined by counting the alternating banding pattern along the dorsal otolith margin. Where ages obtained for an individual differ within these two counts, a third count was performed and the final age was determined by agreement of two separate counts.

Where it was not possible to unequivocally determine placement of the first annuli, or where individuals (probably juveniles or recent recruits) had no clear annual banding pattern, daily growth increments (DGIs) were examined. DGIs were obtained using a compound microscope after successive wet-polishing with 9, 3 and 0.3 µm lapping film, with age in days determined on three separate occasions and the final age was the mean of the three counts. Samples with counts that were outside of 10% of the median were excluded. Parameters of growth were estimated from length-at-age data obtained from all fish collected per site. The von Bertalanffy growth function was used, which is represented by

$$L_t = L_\infty[1 - e^{-k(t-t0)}],$$

in which $L_t$ is the mean predicted fork length (cm) at age $t$ (years) $L_\infty$ is the mean asymptotic fork length, $k$ is the coefficient used to describe the curvature of fish growth towards $L_\infty$ and t0 is the hypothetical age at which FL is equal to zero as described by $k$.

$k$ was standardized relative to the maximum size of *Chromis fieldi* to obtain $K_{max}$ following Morais and Bellwood [40], and using the formula

$$\log_{10}K_{max} = \emptyset + s_L\log_{10}L_{max},$$

in which $K_{max}$ is the expected growth coefficient at the theoretical maximum species size, $\emptyset$ is the growth performance index, $s_L$ is the slope of the relationship between $L_\infty$ (asymptotic fork length) and $k$ (von Bertalanffy growth parameter), and $L_{max}$ is the maximum reported species size.

### 2.3.3. Satellite-derived surface chlorophyll-a concentration

Satellite-derived estimates of nearshore primary production were obtained using chlorophyll-a as a proxy for phytoplankton biomass and availability of planktonic food resources. Remotely sensed surface chlorophyll-a values are well correlated with primary production throughout the photic zone in a range of water types and also well correlated with the abundance of zooplankton [41–44]. We used Ocean Land Colour Instrument (OLCI) imagery from the Sentinel-3A and 3B platforms operated by the European Space Agency, augmented by data from the Moderate Resolution Imaging Spectrometer (MODIS) on the Aqua platform operated by NASA, for the period July 2017 to April 2019. This allowed for overlap of the expected collected fish lifespan (we hypothesized based on the size of fish collected, that many spent approximately 2 years since settlement on reefs). Level 1 OLCI

data were acquired from the NASA Ocean Biology Processing Group (oceancolor.gsfc.nasa.gov) and processed to level 2 using the NASA SeaDAS library function l2gen.

### 2.3.3.1. Satellite data masking procedure

To obtain chlorophyll-a data which were not contaminated with anomalously high chlorophyll-a values associated with altered water column properties in shallow nearshore areas (e.g. bottom reflectance from sand), we created a filtering mask surrounding atolls and removed all pixels inshore of the 30 m isobath *sensu* Gove *et al.* [45] using supervised maximum-likelihood classification of 10 m resolution Sentinel 2 Multi-spectral imager (MSI) data, validated against *in situ* depth sounding data. In the initial Stage 1 submission, we proposed using ETOPO1 bathymetry data [46] to create the 30 m filtering mask, but errors in the ETOPO1 bathymetry were exposed by comparison with *in situ* depth sounding data around islands, and this approach was not used to generate the chlorophyll-a data used in analysis.

### 2.3.3.2. Site-level satellite data procedure

We validated the resulting chlorophyll-a values in offshore areas (depth greater than 100 m and distance from shore greater than 40 km [47,48] against MODIS-Aqua (1 km spatial resolution) chlorophyll-a [49,50] and against those obtained using shallow water optimization with resolved depth (SWORD) values [51]. We generated annual mean chlorophyll-a maps for the entire study period (2017–2019) and produced spatially integrated chlorophyll-a values for each study site. This was achieved using satellite pixel data matching geographic location to the site, but using a 3 × 3 pixel box (300 m OLCI spatial resolution) excluding any depth-masked pixels, giving site level mean and s.d. estimates.

## 2.4. Statistical analysis

To test for associations between the growth and condition of fish and potential variability in deep water nutrient input, we constructed generalized linear models. Models were fitted to each of the following response variables separately: fish growth rate ($K_{max}$), fish condition (C : N ratio) and isotopic value ($\delta^{13}$C and $\delta^{15}$N). All models included atoll (three levels) and collection depth (two levels) as categorical independent variables (fixed factors), together with the continuous independent variables of site chlorophyll-a and reef slope. We examined for interactions between: depth of collection and mean chlorophyll-a, depth of collection and site slope, and mean chlorophyll-a and site slope, for each response variable. The Benjamini–Hochberg method was used to correct for multiple testing, using a false discovery rate of 10% [52].

To further examine whether fish growth rate ($K_{max}$), fish condition (C : N ratio), $\delta^{13}$C and $\delta^{15}$N values differ between collection depths, we used permutation testing to randomly reassign blocks of sites between depths. This tested for a difference in sample means associated with the depth of sample collection (using an alpha value of 0.05) in fish growth rate ($K_{max}$), fish condition (C : N ratio), $\delta^{13}$C and $\delta^{15}$N values.

The specific hypotheses that were tested within this framework, together with the simulated power (from modelPower R function for glm models, and 999 simulations within the emon R package for permutation tests) for each dataset and analysis method, are shown in the design table (table 1).

# 3. Results

We carried out sample data collection and analysis according to the in-principle accepted Stage 1 protocol. The final dataset consisted of size and otolith measurements from 288 *Chromis fieldi* individuals, and paired isotope data from 284 individuals (four samples were damaged during mass spectrometer analysis). Supporting isotopic characterization data consisted of 13 POM samples and 22 pelagic zooplankton samples (https://osf.io/6wjsq/).

## 3.1. Isotopic niche area

The relative trophic position of all taxa was visualized within an isoscape of $\delta^{15}$N and $\delta^{13}$C values (figure 1). Standard Bayesian ellipse areas (SEA_c) fitted showed clear segregation between main trophic groups (POM, zooplankton and fish: figure 1a). There was a small overlap in isotope space between the SEA_c of zooplankton and POM (figure 1a). The SEA_c of fish collected at shallow (10 m)

**Table 1.** Registered report design table listing study hypotheses.

| hypothesis | power analysis | statistical test | interpretation given different outcomes | test result |
|---|---|---|---|---|
| H1 fish growth rate is increased at sites with higher mean chlorophyll-a values | power = 0.81 effect size pEta$^2$ = 0.23 | regression coefficient from generalized linear model | we used the p-value for the chl-a regression coefficient to test for evidence that variation in primary production around atolls influences planktivorous fish growth rate | not supported |
| H2 fish condition (C : N ratio is higher (indicating higher lipid content) at sites with higher mean chlorophyll-a | power = 0.99 effect size pEta$^2$ = 0.58 | regression coefficient from generalized linear model | we used the p-value for the chl-a regression coefficient to test for evidence that variation in primary production around atolls influences planktivorous fish condition | not supported |
| H3 fish nitrogen isotope ($\delta^{15}$N) values depleted at sites with higher mean chlorophyll-a | power = 0.88 effect size pEta$^2$ = 0.3 | regression coefficient from generalized linear model | we used the p-value for the chl-a regression coefficient to test for evidence that variation in primary production around atolls influences planktivorous fish ($\delta^{15}$N) | not supported |
| H4 fish carbon isotope values ($\delta^{13}$C) values will be enriched at sites with higher mean chlorophyll-a | power = 0.99 effect size pEta$^2$ = 0.5 | regression coefficient from generalized linear model | we used the p-value for the chl-a regression coefficient to test for evidence that variation in primary production around atolls influences planktivorous fish ($\delta^{13}$C) | supported |
| H5 fish growth rate is higher at sites with gradual reef slopes facilitating physical delivery mechanisms | power = 0.81 effect size pEta$^2$ = 0.23 | regression coefficient from generalized linear model | we used the p-value for the reef slope regression coefficient to test for evidence that variable delivery of nutrients with reef slope around atolls influences planktivorous fish growth | not supported |
| H6 fish condition (C : N ratio) is increased indicating higher lipid content with gradual reef slopes facilitating physical nutrient delivery mechanisms | power = 0.91 effect size pEta$^2$ = 0.29 | regression coefficient from generalized linear model | we used the p-value for the reef slope regression coefficient to test for evidence that variable delivery of nutrients with reef slope around atolls influences planktivorous fish condition | not supported |
| H7 fish nitrogen isotope ($\delta^{15}$N) will be enriched with gradual reef slopes facilitating physical nutrient delivery mechanisms | power = 0.92 effect size pEta$^2$ = 0.3 | regression coefficient from generalized linear model | we used the p-value for the reef slope regression coefficient to test for evidence that variable delivery of nutrients with reef slope around atolls influences planktivorous fish $\delta^{15}$N | not supported |
| H8 fish carbon isotope values ($\delta^{13}$C) will be depleted with gradual reef slopes facilitating physical nutrient delivery mechanisms | power = 0.99 effect size pEta$^2$ = 0.78 | regression coefficient from generalized linear model | we used the p-value for the reef slope regression coefficient to test for evidence that variable delivery of nutrients with reef slope around atolls influences planktivorous fish ($\delta^{13}$C) | not supported (direction reversed) |
| H9 the relationship between fish growth rate and site chlorophyll-a values is influenced by depth of collection (moderate versus shallow) | power = 0.85 effect size pEta$^2$ = 0.25 | regression coefficient from generalized linear model | we used the p-value for the reef slope regression coefficient to test for evidence that depth influences the relationship between chl-a and fish growth rate | not supported |

(*Continued.*)

**Table 1.** (*Continued*.)

| hypothesis | power analysis | statistical test | interpretation given different outcomes | test result |
|---|---|---|---|---|
| H10 The relationship between fish condition (C : N ratio) and site chlorophyll-a values is influenced by depth of collection (moderate versus shallow) | power = 0.85 <br> effect size pEta$^2$ = 0.25 | regression coefficient from generalized linear model | we used the *p*-value for the regression interaction coefficient to test for evidence that depth influences the relationship between chl-a and fish C : N ratio | supported |
| H11 the relationship between fish nitrogen isotope ($\delta^{15}$N) and site chlorophyll-a values is influenced by depth of collection (moderate versus shallow) | power = 0.85 <br> effect size pEta$^2$ = 0.27 | regression coefficient from generalized linear model | we used the *p*-value for the regression interaction coefficient to test for evidence that depth influences the relationship between chl-a and fish ($\delta^{15}$N) | not supported |
| H12 the relationship between fish ($\delta^{13}$C) and site chlorophyll-a values is influenced by depth of collection (moderate versus shallow) | power = 0.82 <br> effect size pEta$^2$ = 0.25 | regression coefficient from generalized linear model | we used the *p*-value for the regression interaction coefficient to test for evidence that depth influences the relationship between chl-a and fish ($\delta^{13}$C) | not supported |
| H13 the relationship between fish growth rate and reef site slope is influenced by depth of collection (moderate versus shallow) | power = 0.81 <br> effect size pEta$^2$ = 0.23 | regression coefficient from generalized linear model | we used the *p*-value for the regression interaction coefficient to test for evidence that depth influences the relationship between reef slope and growth rate | not supported |
| H14 the relationship between fish condition (C : N ratio and reef site slope is influenced by depth of collection (moderate versus shallow | power = 0.85 <br> effect size pEta$^2$ = 0.25 | regression coefficient from generalized linear model | we used the *p*-value for the regression interaction coefficient to test for evidence that depth influences the relationship between reef slope and C : N ratio | not supported |
| H15 fish nitrogen isotope ($\delta^{15}$N) and reef site slope is influenced by depth (moderate versus shallow) | power = 0.88 <br> effect size pEta$^2$ = 0.29 | regression coefficient from generalized linear model | we used the *p*-value for the regression interaction coefficient to test for evidence that depth influences the relationship between reef slope and ($\delta^{15}$N) | not supported |
| H16 the relationship between fish carbon isotope ($\delta^{13}$C) and reef site slope is influenced by depth (moderate versus shallow) | power = 0.85 <br> effect size pEta$^2$ = 0.25 | regression coefficient from generalized linear model | we used the *p*-value for the regression interaction coefficient to test for evidence that depth influences the relationship between reef slope and ($\delta^{13}$C) | not supported |
| H17 the relationship between fish growth rate and site chlorophyll-a is influenced by reef site slope | power = 0.81 <br> effect size pEta$^2$ = 0.23 | regression coefficient from generalized linear model | we used the *p*-value for the regression interaction coefficient to test for evidence that slope influences the relationship between chl-a and fish growth rate | not supported |
| H18 the relationship between fish condition (C : N ratio and site chlorophyll-a is influenced by reef site slope | power = 0.85 <br> effect size pEta$^2$ = 0.25 | regression coefficient from generalized linear model | we used the *p*-value for the regression interaction coefficient to test for evidence that slope influences the relationship between chl-a and fish C : N ratio | not supported |

(*Continued*.)

**Table 1.** (*Continued.*)

| hypothesis | power analysis | statistical test | interpretation given different outcomes | test result |
|---|---|---|---|---|
| H19 the relationship between fish nitrogen ($\delta^{15}$N) values and site chlorophyll-a is influenced by reef site slope | power = 0.88<br>effect size pEta$^2$ = 0.29 | regression coefficient from generalized linear model | we used the *p*-value for the regression interaction coefficient to test for evidence that slope influences the relationship between chl-a and fish ($\delta^{15}$N) | not supported |
| H20 the relationship between fish ($\delta^{13}$C) values and site chlorophyll-a is influenced by reef site slope | power = 0.85<br>effect size pEta$^2$ = 0.25 | regression coefficient from generalized linear model | we used the *p*-value for the regression interaction coefficient to test for evidence that slope influences the relationship between chl-a and fish ($\delta^{13}$C) | not supported |
| H21 fish growth rate is higher in fish collected at moderate versus shallow depths | power = 0.80<br>effect size *d* = 27%<br>effect size *d* = 0.15 | permutation test between groups ($\alpha$ = 0.05) | we used the *p*-value from permutation testing the probability that the null hypothesis of no difference is true | not supported |
| H22 fish C : N ratio will be higher within fish collected at moderate versus shallow depths | power = 0.83 | permutation test between groups ($\alpha$ = 0.05) | we used the *p*-value from permutation testing the probability that the null hypothesis of no difference is true | not supported |
| H23 fish nitrogen ($\delta^{15}$N) values will be depleted within in fish collected at moderate versus shallow depths | power = 0.80<br>effect size *d* = 0.3 | permutation test between groups ($\alpha$ = 0.05) | we used the *p*-value from permutation testing the probability that the null hypothesis of no difference is true | not supported |
| H24 fish carbon isotope values ($\delta^{13}$C) will be depleted within in fish collected at moderate versus shallow depths | power = 0.82<br>effect size *d* = 0.2 | permutation test between groups ($\alpha$ = 0.05) | we used the *p*-value from permutation testing the probability that the null hypothesis of no difference is true | not supported |

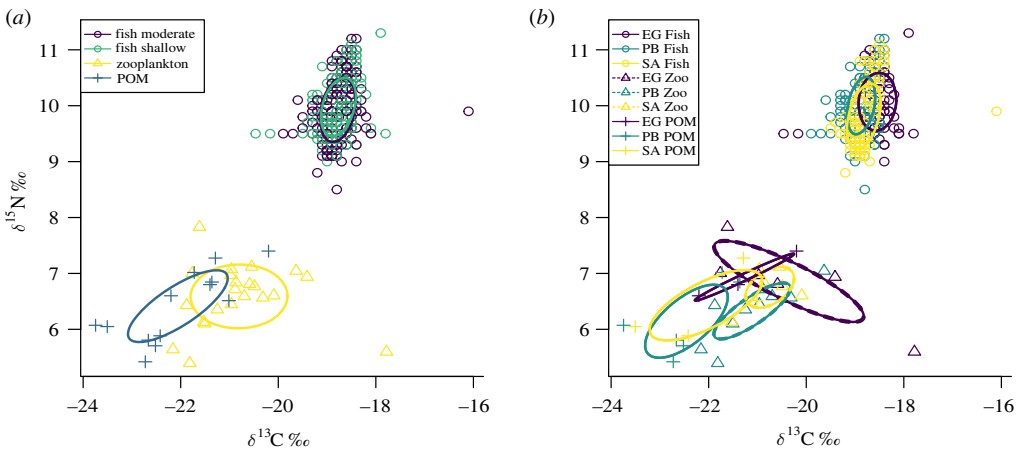

**Figure 1.** Stable isotope biplot of POM, zooplankton and *Chromis fieldi,* (*a*) SEA$_c$ for fish collected at shallow and moderate depth, (*b*) SEA$_c$ for each taxa by atoll.

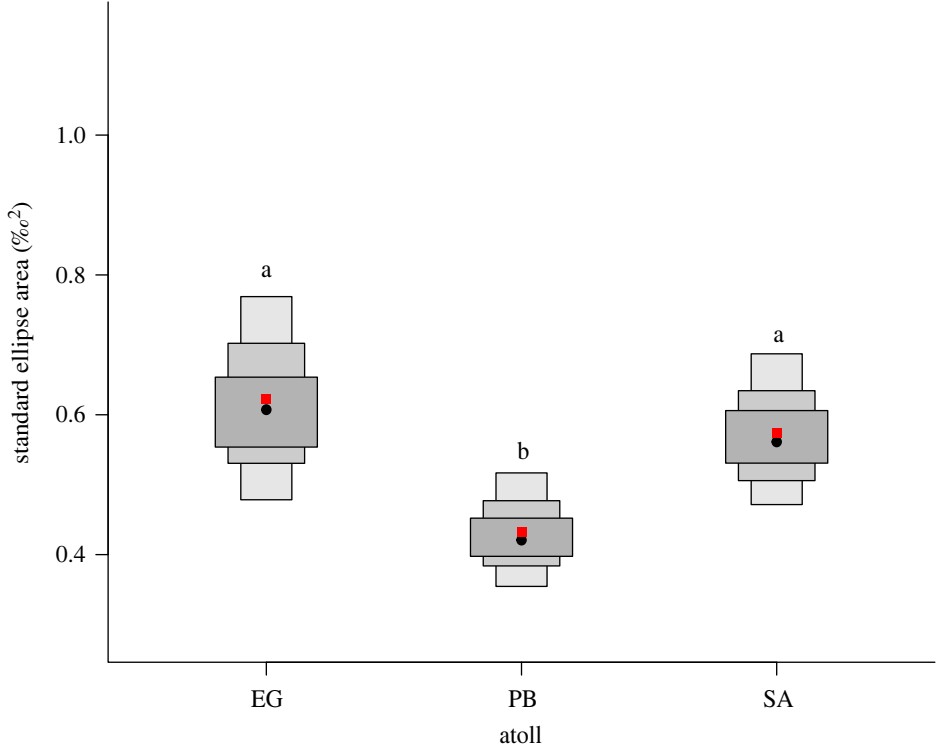

**Figure 2.** Density plots of SEA$_b$ of *Chromis fieldi* by atoll. The population mode is shown by a black dot and boxes of increasing size and colour represent 50%, 75% and 95% credible intervals; the red square represents the SEA$_c$ corrected for sample size according to Jackson *et al.* [38]. Common letters denote no significant difference according to Bayesian inference ($p > 0.05$).

and moderate depths (17.5 m) were closely aligned and occupied a similar position within the isoscape (figure 1*a*). Differences between atolls were apparent in the position of the fish SEA$_c$, mainly along the $\delta^{13}$C axis (figure 1*b*). Differences in the position and size of the zooplankton SEA$_c$ were present between atolls, with the largest SEA at Egmont Atoll (EG) (figure 1*b*). The SEA$_c$ for POM overlapped between Salomon Atoll (SA) and Peros Banhos Atoll (PB), and to a lesser extent for Egmont Atoll (figure 1*b*).

Between atolls, the probability for the posterior distribution (based on 10 000 draws) that planktivorous fish sampled from Egmont Atoll occupied a greater isotopic niche area than Peros Banhos Atoll was 99.2% (figure 2). The probability that fish sampled from Egmont atoll occupied a greater isotopic niche area than from Salomon Atoll was 68%. The probability that fish sampled from Salomon atoll occupied a larger isotopic niche area than from Peros Banhos Atoll was 98% (figure 2).

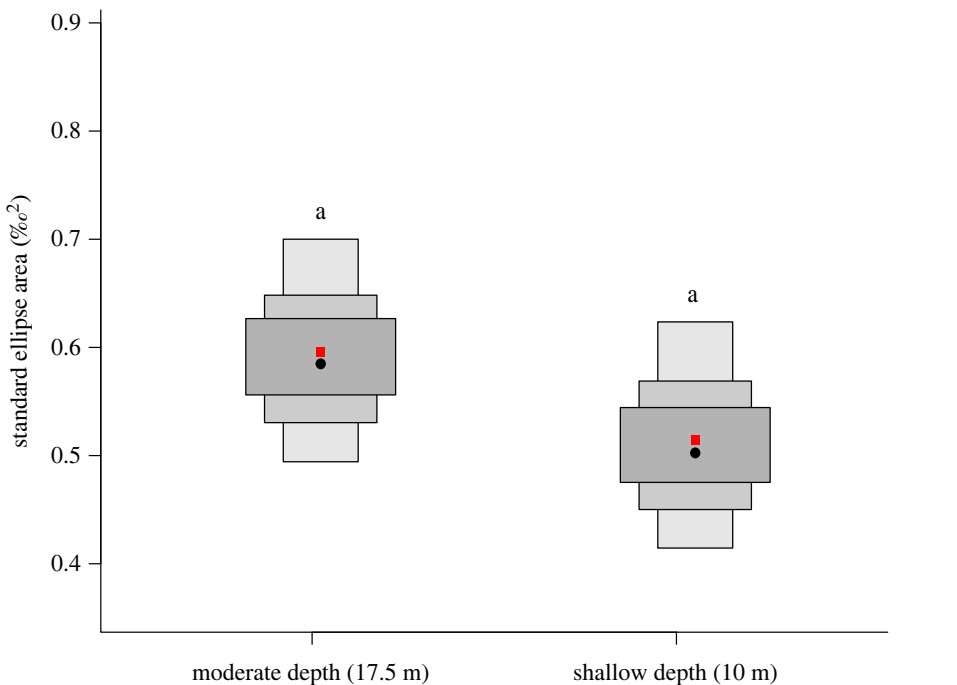

**Figure 3.** Density plots of SEA$_b$ of *Chromis fieldi* by depth. The population mode is shown by a black dot and boxes of increasing size and colour represent 50%, 75% and 95% credible intervals; the red square represents the SEA$_c$ corrected for sample size according to Jackson *et al.* [38]. Common letters denote no significant difference according to Bayesian inference ($p > 0.05$).

Between depths, the probability for the posterior distribution that fish sampled from moderate depths occupied a larger isotopic niche was 87% (figure 3). Comparing depth by atoll, the probability that fish occupied a larger isotopic niche at shallow than at moderate depth was 100% for Egmont Atoll (figure 4). The probability that fish occupied a larger isotopic niche at moderate, relative to shallow depth was 0% for Peros Banhos Atoll, and 99.9% for Salomon Atoll (figure 4).

## 3.2. Fish condition

The metric of fish condition (C : N ratio) obtained from fish muscle tissue samples ranged from 2.9 to 4.7 for each individual. The regression between fish size and condition showed a weak but significant relationship ($R^2 = 0.12$, $F_{1,265} = 36.27$, $p \leq 0.001$; electronic supplementary material, table S2), indicating that as fish size increased, condition decreased.

## 3.3. Fish size and growth rate influence on isotope data

Measurements of total length of *Chromis fieldi* collected ranged from 17 to 70 mm. Site- and depth-specific von Bertalanffy growth curves were obtained and $K_{max}$ standardized growth rate calculated from 27 of a possible total of 36 site-depth combinations (nine site-depth samples had insufficient sample size or no growth curve fit was possible; electronic supplementary material, figures S4–S6). The regression between $K_{max}$ and carbon isotope values was not significant ($R^2 = 0.01$, $F_{1,25} = 0.26$, $p = 0.62$). There was a significant relationship between $K_{max}$ and nitrogen isotope values ($R^2 = 0.48$, $F_{1,25} = 23.26$, $p = < 0.001$; electronic supplementary material, table S3 and figure S7), indicating that as $K_{max}$ increased, $\delta^{15}$N values were depleted.

## 3.4. Satellite-derived chlorophyll-a and reef slope data

Reef slope data were obtained from 16 of 18 *in situ* stations and reef slope angles ranged from −16 to −70 degrees (https://osf.io/6wjsq/). Sentinel-3A/B OLCI surface chlorophyll-a data were obtained from 3 × 3 pixel boxes (300 m spatial resolution) for 16 of 18 *in situ* stations and were time-averaged over the period July 2017 to August 2019 (figure 5a). Results of the masking procedure using supervised maximum-likelihood habitat classification of Sentinel MSI data validated with *in situ* reef profiles are

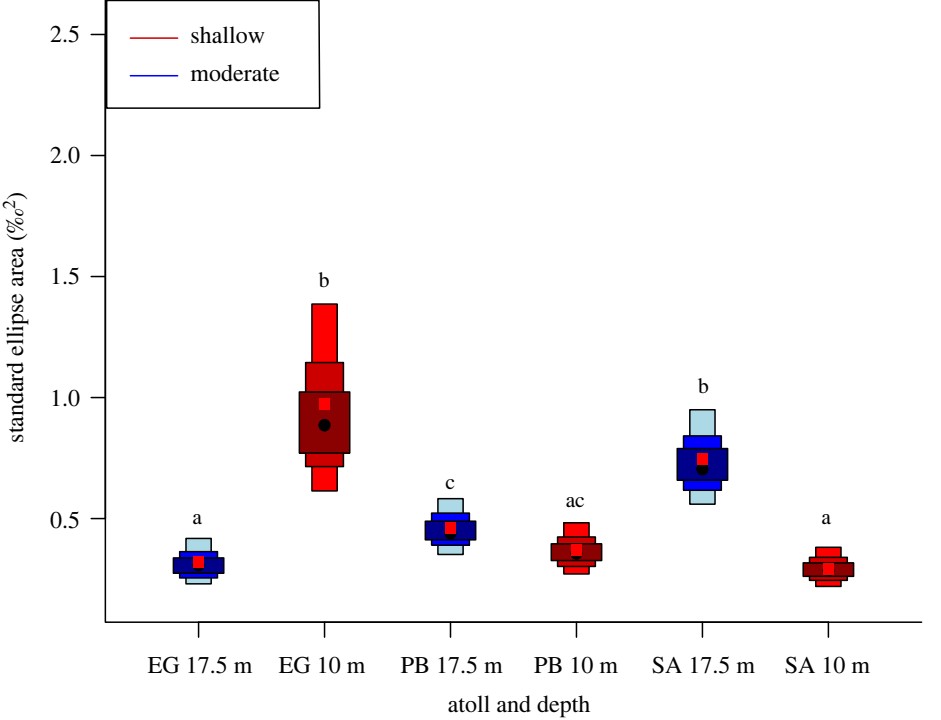

**Figure 4.** Density plots of SEA$_b$ of *Chromis fieldi* by atoll and depth. The population mode is shown by a black dot and boxes of increasing size and colour represent 50%, 75% and 95% credible intervals; the red square represents the SEA$_c$ corrected for sample size according to Jackson *et al*. [38]. Common letters denote no significant difference according to Bayesian inference ($p > 0.05$).

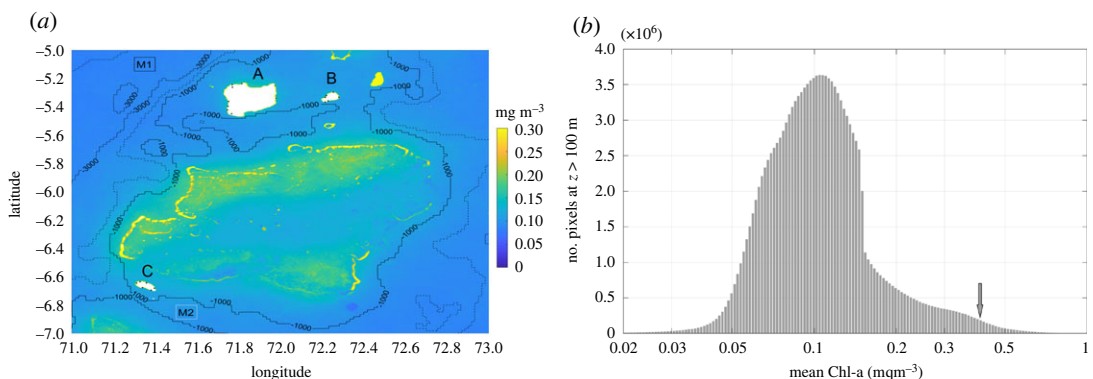

**Figure 5.** (*a*) Time-averaged (July 2017–August 2019) chlorophyll-a from Sentinel 2 and 3 OLCI. Depth contours are taken from ETOPO-1 [46]. Boxes M1 and M2 are the regions in greater than 1000 m depth chosen for comparison against MODIS-Aqua. Atolls (A) Peros Banhos, (B) Salomon and (C) Egmont are masked using Sentinel 2 MSI masks. (*b*) Frequency histogram of time-averaged chlorophyll-a (excluding areas shallower than 100 m) for the study domain shown within (*a*). The value of the highest chlorophyll-a site sampled at Egmont Atoll is indicated by an arrow within this histogram.

shown for Egmont Atoll (figure 6) and Salomon and Perhos Banhos Atolls (electronic supplementary material, figures S9 and S10). Mean chlorophyll-a across all 16 sites was $0.148 \pm 0.068$ mg m$^{-3}$. Mean chlorophyll-a values were higher at sites located around Egmont Atoll ($0.211 \pm 0.12$ mg m$^{-3}$) than Salomon Atoll ($0.129 \pm 0.006$ mg m$^{-3}$) and Peros Banhos Atoll ($0.123 \pm 0.004$ mg m$^{-3}$). Across all three atolls, site chlorophyll-a ranged from 0.116 to 0.405 mg m$^{-3}$. The highest site chlorophyll-a values from the northern section of Egmont Atoll ('Egmont Mid': 0.405 mg m$^{-3}$) were at least two times greater than all other sites; however, satellite data processing showed no indication that this site was more likely to be influenced by issues of reef glint or bottom reflectance than others within the dataset and found consistently high temporal variability at this site (electronic supplementary material, figure S17). A histogram of time-averaged chlorophyll-a values within the study domain at

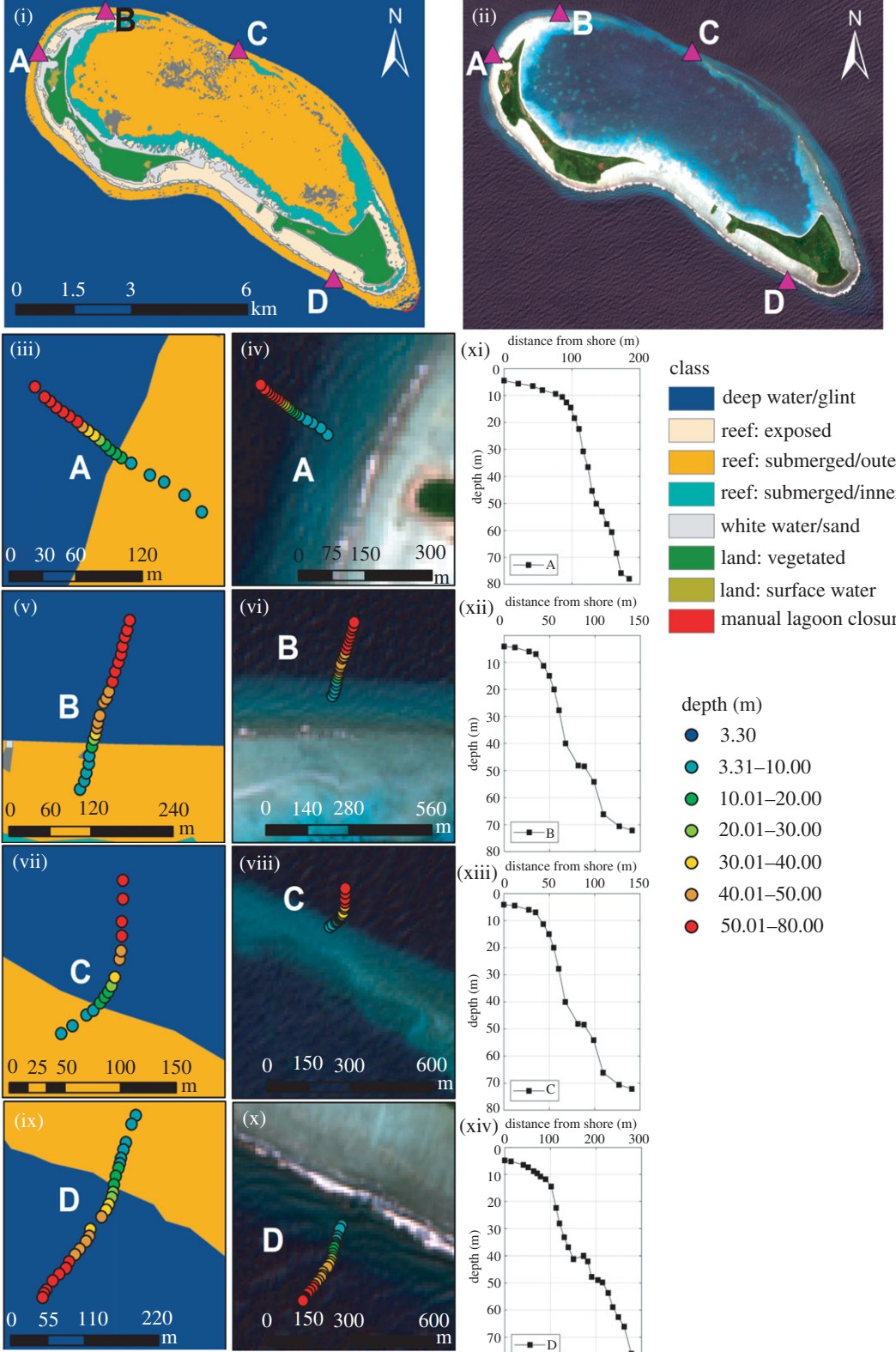

**Figure 6.** (i) Habitat masking map produced from Sentinel 2 MSI data for Egmont Atoll, (ii) reef slope site locations (A, B, C, D) around Egmont Atoll, (iii, v, vii, ix) sites A, B, C, D *in situ* depth soundings plotted on habitat masking map, (iv, vi, viii, x) sites A, B, C, D *in situ* depth soundings plotted on Sentinel satellite imagery, (xi, xii, xiii, xiv) reef profiles obtained from *in situ* depth soundings of sites A, B, C, D.

300 m spatial resolution (excluding areas shallower than 100 m), showed values were mainly distributed between 0.05 and 0.17 mg m$^{-3}$ (figure 5b), with a long tail of higher chlorophyll-a values extending from 0.17 to greater than 0.5 mg m$^{-3}$ (figure 5b).

**Table 2.** Summary of generalized linear model results. Values in parentheses are coefficient standard errors.

| | dependent variable: | | | |
|---|---|---|---|---|
| | dN | dC | CN | $K_{max}$ |
| | normal | normal | normal | gamma |
| | nitrogen | carbon | C : N ratio | $K_{max}$ |
| | (1) | (2) | (3) | (4) |
| constant | 9.831*** | −18.622*** | 3.259*** | 0.733*** |
| | (0.284) | (0.081) | (0.033) | (0.211) |
| depth | 0.086 | 0.039 | 0.001 | −0.038 |
| | (0.137) | (0.039) | (0.016) | (0.098) |
| slope | −0.771 | −0.503*** | −0.044 | −0.292 |
| | (0.587) | (0.167) | (0.067) | (0.466) |
| Chl-a | −9.446 | −5.289** | −0.204 | −3.727 |
| | (6.693) | (1.905) | (0.768) | (5.494) |
| depth : slope | −0.120 | 0.032 | 0.037 | −0.007 |
| | (0.231) | (0.066) | (0.026) | (0.261) |
| depth : Chl-a | −1.690 | −1.340 | 1.104*** | 1.069 |
| | (2.007) | (0.571) | (0.230) | (1.431) |
| slope : Chl-a | −38.602 | −20.149** | −0.352 | −15.214 |
| | (25.279) | (7.195) | (2.900) | (20.688) |
| observations | 30 | 30 | 30 | 26 |
| log likelihood | −8.791 | 28.906 | 56.165 | 1.089 |
| Akaike inf. crit. | 35.582 | −39.811 | −94.330 | 15.821 |

Note: *$p < 0.1$; **$p < 0.05$; ***$p < 0.01$.

## 3.5. Primary production

Hypotheses 1 to 4 tested for relationships between primary production (chlorophyll-a), and fish isotopic composition, condition and growth (table 1). Fish carbon isotope $\delta^{13}C$ values were depleted at sites with higher mean chlorophyll-a values indicated by a significant negative effect of the chlorophyll-a variable within the carbon isotope linear regression model ($B = −5.29$, s.e. $= 1.90$, $t = −2.78$, $p = 0.011$; table 2). Hypotheses 1 to 3 were unsupported by the data, and hypothesis 4 was supported (table 1).

## 3.6. Reef slope

Hypotheses 5 to 8 tested for relationships between reef slope and fish isotopic composition, condition and growth. Hypotheses 17 to 20 tested for interactions between reef slope, chlorophyll-a and fish isotopic composition, condition and growth (table 1). The relationship between fish carbon isotope $\delta^{13}C$ and reef site chlorophyll-a was influenced by reef site slope, indicated by a significant interaction effect within the carbon isotope regression model ($B = −20.15$, s.e. $= 7.19$, $t = −2.8$, $p = 0.011$; figure 7). Fish carbon isotope $\delta^{13}C$ values were depleted with increasing reef slope, indicated by a significant effect of slope within the carbon isotope regression model (table 2). The direction of this effect was reversed compared with the hypothesis of depleted $\delta^{13}C$ values with gradual slopes (table 1). Hypotheses 5 to 8, and 17 to 19 were unsupported by the data and hypothesis 20 was supported (table 1).

## 3.7. Depth

Hypotheses 9 to 16, and 21 to 24 tested for relationships between depth and fish isotopic composition, condition and growth (table 1). The relationship between fish condition and site chlorophyll-a was

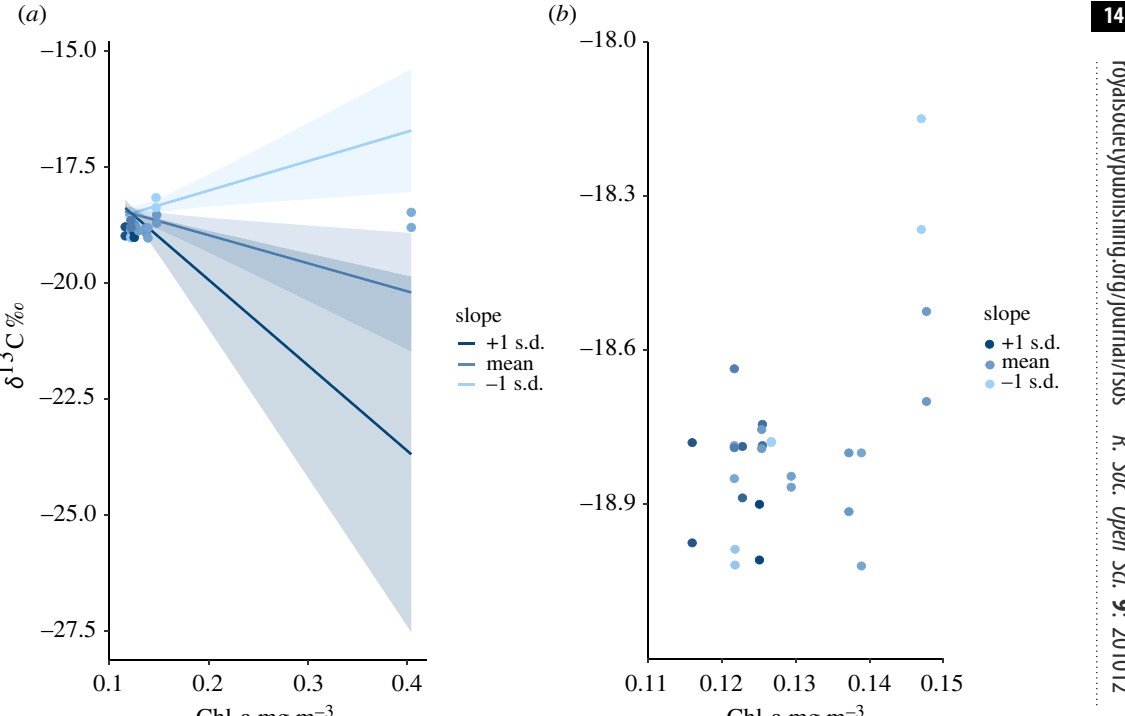

**Figure 7.** Relationships between planktivorous damselfish carbon isotope $\delta^{13}$C and satellite-derived reef site chlorophyll-a values influenced by steepness of reef slope (a) using all sites within the dataset and (b) removing the highest site-level mean chlorophyll-a value.

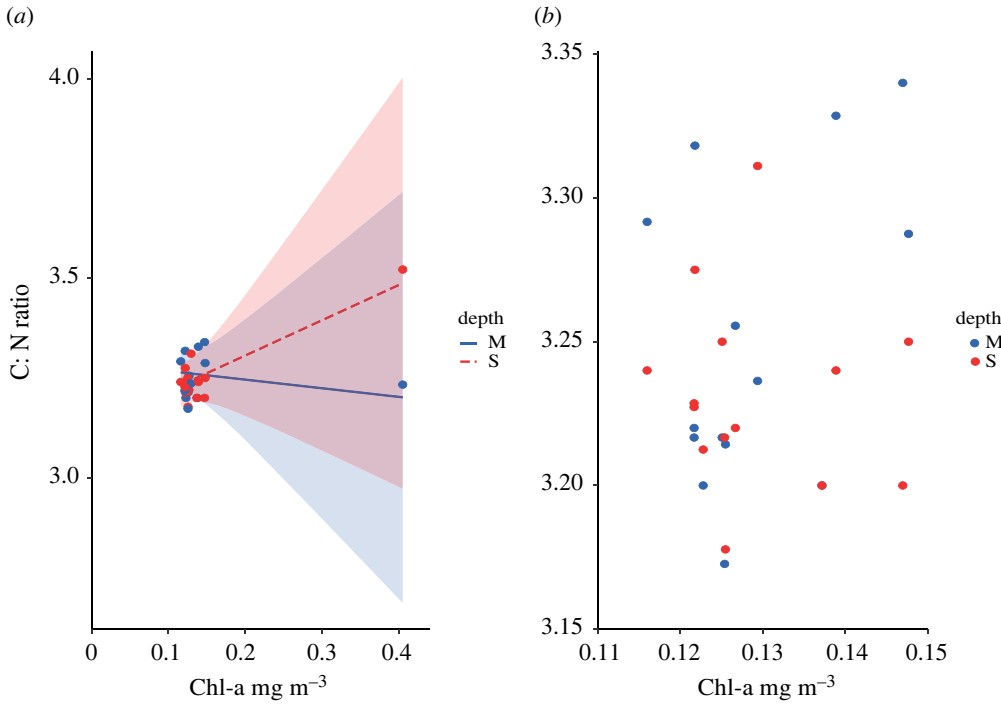

**Figure 8.** Relationships between planktivorous damselfish condition (C : N ratio) and satellite-derived reef site chlorophyll-a values influenced by the depth of collection: shallow (10 m) versus moderate (17.5 m), (a) using all sites within the dataset and (b) removing the highest site-level mean chlorophyll-a value.

influenced by the depth of collection, indicated by a significant interaction effect within the fish condition (C : N) regression model ($B = 1.11$, s.e. $= 0.23$, $t = 4.8$, $p \leq 0.001$; figure 8). Hypothesis 9, and 11 to 16 were unsupported by the data (table 1). Hypothesis 10 was supported (table 1). Hypotheses 21 to 24 were

unsupported. There were no significant differences in mean values of fish growth rate, fish condition, fish nitrogen isotope values or fish carbon isotope values when data from shallow and moderate depth collection sites were pooled together and permuted 9999 times between depths (https://osf.io/6wjsq/).

## 4. Discussion

We examined the effect of depth, variation in primary production and reef slope on $\delta^{15}N$, $\delta^{13}C$, growth rate and condition of planktivorous fish across 18 sites in the central Indian Ocean. We found no difference in mean isotopic value, growth rate or condition between fish grouped by depth of collection. There was an interactive effect of depth and primary production on fish condition, and an interactive effect of reef slope and primary production on fish $\delta^{13}C$ values. These interactions were driven by high primary production values present at one study site. In the paragraphs below, we first discuss the absence of an effect of depth on fish growth and condition. Second, we examine the interactive effects of primary production, and the distribution of chlorophyll-a values within the study domain. Third, we interpret relationships between fish size and condition, and the effect of growth rate on fish isotope values. Fourth, we draw inferences from isotopic niche area analysis. Finally, we discuss limitations of the present study and suggest data collection methods to strengthen further research on the effects of deep water nutrient inputs to coral reef planktonic food chains.

We hypothesized that planktivorous fish collected at greater depths would exhibit enriched $\delta^{15}N$ values, consistent with previous findings in the central Pacific [24] and Western Australia [31]. Contrary to our prediction, we did not find a simple depth-dependent pattern within fish $\delta^{15}N$ values, which suggests that site-specific processes and characteristics unrelated to depth influence fish $\delta^{15}N$. Alternatively the depth range sampled (10–17.5 m) was insufficient to detect an effect of deepwater nutrient inputs. Our original hypothesis assumes that the zooplankton communities which planktivorous fish consume differ in $\delta^{15}N$ between depths due to proximity to nutrient delivery from deep water oceanic sources. There are several instrumental datasets, which, while limited in spatial and temporal resolution, suggest movement of colder water from depth to shallow reefs within the Chagos Archipelago [53,54]. Therefore, rather than an absence of these processes across the study sites, the most likely explanation for the lack of a $\delta^{15}N$ pattern is that zooplankton are influenced by a common oceanic $\delta^{15}N$ source across sites and that the water column and zooplankton communities are well-mixed across sampled depths. A lack of spatial difference in $\delta^{15}N$ within planktivorous fish sampled between locations with varying oceanic influence has previously been observed and attributed to similarity in food chain length between locations [25,28].

We also found no significant effect of depth, primary production or reef slope on fish growth rates. This indicates *Chromis fieldi* growth at these sites is not limited by zooplankton availability and that gradients in zooplankton abundance, or variable reef bathymetry between sites did not translate into fish growth rate differences. An increase in the abundance of planktivorous fishes, peaking at moderate (20–30 m) and mesophotic (40–70 m) depths has been widely documented [29,55,56], but evidence for associated growth rate differences across depth gradients on coral reefs is lacking. Contrary to our hypothesis of increasing fish growth rate with proximity to deep water nutrient sources, slower growth trajectories have been found in planktivorous damselfish collected at mesophotic (60–70 m), relative to shallow depths (10 m) at sites in the Florida Keys [57]. The absence of growth rate differences within our study species suggests a lack of response to primary production gradients. Similar spatially invariable growth patterns in spite of strong gradients of chlorophyll-a (mean chlorophyll-a of 0.79 mg m$^{-3}$ within the Gulf of Aden versus 0.16 mg m$^{-3}$ in the Northern Red Sea) occur within butterflyfish species across the Red Sea region [58].

There was an interactive effect of depth and primary production on fish condition (C:N ratio). At sites with higher primary production, fish condition was higher at shallower depths (figure 8a). The higher fish C:N ratios recorded within shallow depths at high primary productivity sites were probably driven by fine-scale oceanographic processes occurring at one of the three study atolls (Egmont Atoll), which contained sites with the highest mean chlorophyll-a values. Indeed, when the site with notably high chlorophyll-a is removed, there is no significant relationship between C:N ratio, chlorophyll-a values and depth (figure 8b). Overall, this indicates that planktivorous fish condition does not vary across the range of chlorophyll-a values (0.11–0.147 mg m$^{-3}$) in the remaining study sites, which fall within the most frequent mean chlorophyll-a concentrations in the study domain (figure 5b).

The implications of these findings are notable on two levels. First and most specifically, the sites surveyed at Egmont Atoll appear anomalous and worthy of further investigation. This atoll has a partially enclosed lagoon, which is connected to the adjacent reef slope by a shallow (approx. 5 m) permanently submerged atoll rim along its northern margin. High localized zooplankton abundances have been detected at this location using acoustic backscatter [54]. These have been attributed to a process in which cold water bores transport zooplankton from depth (greater than 50 m) and into the lagoon on flood tides, where they are aggregated before being pumped out across the rim at shallow depths [54]. This is consistent with the higher planktivorous fish condition found at shallow depths on the northern section of the atoll. Second, and more generally, here we sampled sites within a broad but right-skewed distribution of chlorophyll-a values and found no evidence of a mechanistic link between high primary production and planktivorous fish growth or condition, only potentially on the extreme end. This suggests that using the Chagos Archipelago to establish 'baseline or benchmark' values for Indian Ocean coral reef fish biomass [59–61] may not capture the full influence of primary production gradients on fish populations unless sampling site selection for baseline estimates are stratified by this distribution [62].

Fish $\delta^{13}C$ isotope values were depleted at sites with higher mean chlorophyll-a values, consistent with expectations based on a reliance on primary production derived from deeper oceanic nutrient sources. This difference in $\delta^{13}C$ with higher mean chlorophyll-a was influenced by an interaction with reef slope (figure 7a). At mean reef slope values, and at reef slopes higher than the mean, fish $\delta^{13}C$ was depleted with increasing primary production. At shallow reef slopes below the mean, $\delta^{13}C$ isotope values were enriched with increasing primary production. While influenced by the high chlorophyll-a values present at the Egmont Atoll site, the relationship remains if this site is removed from the dataset (figure 7b). The interaction between reef slope and chlorophyll-a suggests that planktivore $\delta^{13}C$ values are driven by site-specific characteristics across this study domain. Isotopically distinct planktonic carbon pathways (nearshore reef-associated plankton and offshore pelagic plankton) identified using compound-specific approaches [63], illustrate the complexity underlying bulk fish $\delta^{13}C$ as obtained here, and the potential importance of interactions between reef slope and hydrodynamic processes affecting the strength of co-occurring carbon isotope pathways.

We found a weak negative relationship between fish size and condition, indicating that smaller fish tended to have higher condition (electronic supplementary material, table S2). This suggests that while fish size is not a strong predictor of condition, larval history and lipid content increases prior to metamorphosis may translate to higher condition among smaller juvenile fishes [64–66]. We found a growth rate effect on fish $\delta^{15}N$ values, but no relationship between growth rate and $\delta^{13}C$ values (electronic supplementary material, table S3). This pattern is consistent with controlled dietary studies where fish growth rate has been altered within captive Atlantic salmon (*Salmo salar*) and summer flounder (*Paralichthys dentatus*) which found depletion in $\delta^{15}N$ values as growth rates increased [67,68]. The most likely explanation for this pattern is that increasing fish growth rate is accompanied by greater nitrogen use efficiency, influencing fish tissue $\delta^{15}N$ values [67]. This finding suggests that $\delta^{15}N$ variation in *Chromis fieldi* is driven primarily by metabolic processes and growth rate changes, rather than coral reef primary production gradients.

Isotopic niche area provides information on the trophic niche occupied by a species [38]. In general, increases in system productivity will result in a more trophically complex ecosystem, thus increasing trophic niche size [69]. This pattern has been observed within planktivorous fish across a strong north–south gradient of oceanic primary production in the Southern Line Islands, where trophic niche expanded with increasing nearshore production [70]. While larger at moderate depths, we found differences in isotopic niche area between depths were not statistically significant (figure 3). By contrast, significant differences in isotopic niche size were observed when samples were grouped by atoll, providing evidence of spatial variation in the ecological niche occupied by planktivorous fish at scales of greater than 150 km (figure 4). The larger isotopic niche at shallow depths within Egmont Atoll (figure 4) is consistent with bathymetry and fine-scale oceanographic processes concentrating zooplankton towards the surface at this location, in a reversal of the general depth pattern. Similar context-specific oceanographic regimes which reverse or homogenize expected trophic zonation across depths have previously been recorded in the central Pacific [24].

The limitations to this study include the restricted depth range of sampling due to diving safety restrictions in a remote location. Goldstein *et al*. [29] examined damselfish growth between shallow (less than 10 m), deep shelf (20–30 m) and mesophotic (60–70 m) reef sites in the Florida Keys, and found significant differences only between shallow and mesophotic depths. Although we characterized surface zooplankton communities across sites, targeted sampling of zooplankton within

specific depth ranges would facilitate testing the hypothesis that these communities differ with increasing proximity to deep water nutrient sources. Finally, although we used the highest resolution of publicly available satellite data to derive chlorophyll-a values as a proxy of primary production within the photic zone, we currently lack detailed site-level physical data recording water movements and nutrient concentration across depths to quantify the intensity of processes transporting nutrient-enriched water onto reef slopes. These data would allow a more accurate analysis of the influence of allochthonous nutrient sources on reef fish community growth and condition, and how these vary seasonally within this monsoon-dominated system.

The predicted relationship of higher fish growth rate and depleted $\delta^{15}$N values with increasing depth was not observed within our dataset, indicating that any influence of deep water nutrient input on these variables is not detectable in the planktivorous fish *Chromis fieldi* in the shallow to moderate depths we examined (10–17.5 m). Nonetheless, we provide evidence that sites which are 'hotspots' of high primary production can influence relationships with planktivore isotopic composition ($\delta^{13}$C) and condition (C : N ratio), and we found that planktivore trophic niche area differs between shallow and moderate depths within atoll reef systems. While the importance of energetic contributions from external sources to coral reef ecosystems is increasingly evident [5,7,62,63,71], our results show how expected broad-scale patterns can be altered by site-specific variation in physical characteristics influencing planktonic food chains.

Ethics. Ethical approval to conduct fish sampling and euthanasia was given by the ethics committee of the College of Science and Engineering Bangor University. A permit to carry out research was provided by the British Indian Ocean Territory Administration (Ref: BPMS Reef 2 Project 2 2019).

Data accessibility. The pre-registered Stage 1 report can be found on the Open Science Framework at the following link: https://osf.io/bgqxk/. The code for data analysis and datasets used can be found in the OSF project link: https://osf.io/6wjsq/. The raw isotope and otolith growth datasets have been submitted to the Dryad Digital Repository, at the following link: https://doi.org/10.5061/dryad.q83bk3jkt.

The data are provided in the electronic supplementary material [72].

Authors' contributions. R.C.R.: formal analysis, investigation, writing—original draft and writing—review and editing; A.H.: conceptualization and writing—review and editing; B.M.T.: investigation and writing—review and editing; J.N.S.: formal analysis and writing—review and editing; M.D.F.: conceptualization and writing—review and editing; L.K.S.: investigation; G.J.W.: conceptualization and writing—review and editing; J.R.T.: funding acquisition and project administration.

All authors gave final approval for publication and agreed to be held accountable for the work performed therein.

Conflict of interest declaration. We declare we have no competing interests.

Funding. Fieldwork was supported by the Fondation Bertarelli.

Acknowledgements. We thank the captain and crew of the Grampian Frontier for their assistance during fieldwork. We also acknowledge the support given to the expedition by the British Indian Ocean Territory Administration. We would like to thank Susan Allender for sampling advice and laboratory assistance.

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
