## [Peer Review File · Royal Society Open Science]

Review History

Decision letter (RSOS-200319.R0)

04-Mar-2020

Dear Dr Roche,

I write you in regards to manuscript RSOS-200319 entitled "Linking variation in planktonic primary production to coral reef fish growth and condition" which you submitted to Royal Society Open Science.

We routinely triage submissions for scientific soundness, clarity and general adherence to the Registered Reports guidelines. For submissions that have promise but are not yet suitable for in-depth Stage 1 review, we offer feedback to help authors maximise the chances that reviewers will respond positively to a resubmission.

We have concluded that your submission is not yet suitable for in-depth review and has therefore been rejected at this time, but we believe it will be suitable once several issues are addressed. We therefore invite a resubmission. Further comments from the Associate Editor may be found at the end of this letter.

If you wish to revise your manuscript in light of the below comments please submit your manuscript as a new submission and mention this previous manuscript ID in your covering letter. You should also provide a detailed response to the below comments in the cover letter.

Please note that Royal Society Open Science will introduce article processing charges for all new submissions received from 1 January 2018. Registered Reports submitted and accepted after this date will ONLY be subject to a charge if they subsequently progress to and are accepted as Stage 2 Registered Reports. If your manuscript is submitted and accepted for publication after 1 January 2018 (i.e. as a full Stage 2 Registered Report), you will be asked to pay the article processing charge, unless you request a waiver and this is approved by Royal Society Publishing. You can find out more about the charges at <https://royalsocietypublishing.org/rsos/charges>. Should you have any queries, please contact openscience@royalsociety.org.

Thank you for considering Royal Society Open Science for the publication of your registered report.

on behalf of Professor Chris Chambers (Registered Reports Editor, Royal Society Open Science)
openscience@royalsociety.org

Associate Editor Comments to Author:

Comments to the Author:

The proposal is promising but needs some refinement to provide additional clarity and methodological detail. Please address the following points in a resubmission:

1. Make clear the total sample size that will be subjected to statistical analysis using linear mixed models. This was not immediately apparent from the Method.
2. Consider a prospective power analysis or justify the absence of a formal sampling plan. For linear mixed models, simulation methods may be the most appropriate approach for calculating power (eg. <https://besjournals.onlinelibrary.wiley.com/doi/pdf/10.1111/2041-210X.12504>) but the authors are free to choose an approach to their liking. For frequentist statistical methods, reviewers are likely to expect some formal consideration of power during Stage 1 RR assessment. Even though the sample size is likely to be fixed (given that the data are already collected), you could report a sensitivity power analysis which calculates the effect size that your design has reasonable power (e.g. 80% or 90%) to detect given the available sample size.
3. Ensure a direct and precise correspondence between the hypotheses, the critical statistical tests or test components that will test those hypotheses, if appropriate the statistical power of each test or test component, and the interpretation given different outcomes. To ensure maximum clarity please include in the Method section a design table as shown in section 9 of this template (<https://osf.io/93znh/>) outlining the research question, hypothesis, sampling plan, analysis plan, and which outcomes will lead to which interpretation. Attached to this letter is a PDF file containing some examples from previous submissions of how these tables can look (from

different research fields to the current proposal, but the principles are the same for all statistical hypothesis testing)

4. The abstract includes (in yellow highlight) two potential summaries of the results. This is an admirable commitment to prespecification, however it seems possible that the results could be more complex and may eventually require a different statement altogether. Therefore, it is recommended that the authors replace "will" with "may" in the accompanying note: "Depending on the results obtained sections from the either the second or the third portion of the abstract highlighted **may** be used as appropriate."

Author's Response to Decision Letter for (RSOS-200319.R0)

See Appendix A.

RSOS-201012.R0

Review form: Reviewer 1

Do you have any ethical concerns with this paper?

No

Recommendation?

Accept with minor revision

Comments to the Author(s)

The research question addresses the important and timely question of whether/how spatial differences in primary productivity (nutritional resources) affect fish populations. There is great interest in the relationship between primary productivity and coral reef ecosystem function, including fish growth and its contribution to biomass accumulation. Similar questions have been investigated at a global scale across fish of different dietary categories (e.g., Morais and Bellwood 2018) but the scales at which primary productivity was considered presented challenges for investigating the response of fish growth. This proposal reviewed here presents a unique research question that would allow for an in-depth examination of specific growth parameters of one species of fish across three atolls (18 sites, I believe) of an oceanic atoll system. The question is approached in a logical manner, with a thorough breakdown providing details on each of the hypotheses to be tested. The design is appropriate and considers several different important environmental aspects that may have a role in nutrient delivery across the reef slope. I recommend that this Stage 1 manuscript be accepted following minor revisions.

Additional brief detail about the focal species would be welcome (e.g., reef-associated planktivore; how far off the reef does it typically feed?).

Ideally, the project would examine $\delta^{13}\text{C}$ and $\delta^{15}\text{N}$ values of expected food sources in the water column where the fish were collected. See comment below about the inclusion of zooplankton and particulate organic matter samples, as stated in the Methods of the original submission of the manuscript.

Other potential positive controls to have an idea of baseline isotope values (but of course does not account for fractionation, which is why zooplankton samples would be better) would be to consider whether the group/collaborators have access to any specimens of benthic reef organisms that have long turn-over time for tissue C and N. I am aware that there are d15N measurements of turf algae from the reef crest from Chagos (Graham et al. 2018), but perhaps the authors know of other papers or collaborators who may have measured stable isotope ratios in benthic organisms.

Additional comments:

Please use continuous line numbering throughout the document rather than page-by-page line numbers to assist the reviewers and editors. Also, the line numbers do not appear to match up to the actual lines for some reason, so the following comments related to line numbers may be slightly off.

Abstract:

Lines 26-27: “fish growth and condition will be higher” perhaps needs to be revised to something along the lines of “parameters of fish growth and condition will be higher” otherwise “fish... condition will be higher” is unclear (because condition has not yet been defined).

Lines 43/44: Spelling - “there [were] no significant relationships...”

Lines 46-50: “...no difference in d15N or d13C values in fish collected from sites with higher mean chlorophyll-a, or at greater depths across reefs, indicating a homogeneous primary production resource” does not necessarily “suggest an absence of deep-water oceanic nutrient influence”. It is possible that mixing distributes the deep-water oceanic nutrient source so that differences are not detected between the depths sampled, which are not drastically different in space (17.5 m and 10 m). It is also possible that the sampling depths simply were not deep enough to detect a potential influence of deep-water oceanic nutrient source. This should be re-written more openly so as not to dismiss the other reasons why a deep-water nutrient source was not detected, and then caveats discussed in the Discussion.

Introduction:

I was surprised not to see any mention of Letessier et al. 2016 (Enhanced pelagic biomass around coral atolls) in the Introduction as this research not only is focused on pelagic biomass and fish biomass but also presents data from Chagos (same location as this proposal). Given the close ties and relevancy to the objectives of this paper, it'd be ideal to tie this work into your Introduction.

Samoilys et al. 2018 (Patterns in reef fish assemblages: Insights from the Chagos Archipelago) presents fish biomass data (total biomass, as well as planktivore biomass) for several of the same Chagos atolls (including fore reef sites) and similar depth range that you consider in your study. This paper shows that planktivore biomass differs significantly between atolls in Chagos. Given the relevance of this, this could likely be incorporated into your Introduction as you set the scene for your question of whether fish growth rates differ across the archipelago.

It is worth mentioning in the Introduction that you will be focusing on planktivorous fish, such as in the last paragraph with the hypotheses.

Page 3, lines 9-16: The sentence “This productivity paradox...main mechanisms driving productivity” should be rewritten for clarity. It is worth to consider the wording “This productivity paradox has predominantly been explained by internal reef processes” (with the

concern here being “internal reef processes”) because hydrodynamic processes have a critical role in both providing nutrients and facilitating nutrient exchange.

Page 4, lines 20-21: “...in response to variability in their energy environment...” – what do you mean by “energy environment”?

Methods:

Considering the dominant southeast winds in Chagos, you may also consider the exposure of your different sites across the atolls (e.g., sheltered versus exposed to the dominant southeast winds). For example, see Perry et al. 2015 (“Remote coral reefs can sustain high growth potential and may match future sea-level trends”) characterization of wave exposure across some of the same atolls that you examine in your study. Perhaps these exposure differences will be clear among your site mean chlorophyll-a values?

You likely may find that remote sensing data is quite patchy during the wet monsoon season. To this end, you may consider generating mean chlorophyll-a maps for a time period longer than only 2017-2019 if missing data affects the analysis.

Lines 12-14: When did you collect the fish? More information is needed – what were the sampling dates? The date of collection is important in relation to the environment experienced by the fish prior to the collection (as noted in the Introduction, delivery of nutrient subsidies are affected by temporal variability). Also, the reader will want to know whether the fish were all collected in one time period, as this is important to the interpretation of the stable isotope data.

Lines 31-38: The depth recordings used to obtain information about the physical steepness of the reef slope is very interesting, and this method could be of interest to the community as reef slope parameters are not recorded often enough. Please provide further details about the method used.

Proposed Analyses:

In the original submission (RSOS-200319), page 6 lines 47-58 and page 7 lines 29-34, stated that both particulate organic matter and zooplankton were collected in the field and that you planned to complete stable isotope analyses for both of these sources. This is important in situ data in relation to the satellite-derived data. Why was this removed from the revised submission? This would be very useful data if available.

Perhaps the authors may consider standardizing K relative to the maximum size of the focal species instead of L_{∞} , as argued for and demonstrated by Morais and Bellwood 2018 (Global drivers of reef fish growth). At the least, it is worth checking whether or not L_{∞} exceeds the reported maximum size of the species by 50% or more.

Lines 8-9: Typo (“...a tin capsules”).

Line 27: The variable “ R ” (e.g., R_{sample}) is not defined for the formula presented.

Lines 48-51: You may want to add the additional detail that ... “the linear relationship has been established between C:N and lipid content [for aquatic animals]”.

Proposed Analyses, Page 3:

Line 33: Specify the spatial resolution of the MODIS Aqua chlorophyll-a data (4 km?). In Figure 1 of the original submission (RSOS-200319), some sites (per atoll) appear quite close together so it is

important to state the resolution of the chlorophyll-a data (as well as other remote sensing products mentioned) and add details about how the proximity of sites will be treated in relation to assigning mean chlorophyll-a values per site.

Statistical analysis, page 9:

Lines 38-39: Due to predominant southeast wind and therefore wave exposure, it would make more sense to include individual sites as a random effect instead of the atoll/island level which encompasses sites across all sides of each atoll. It also makes more sense to have site as a random effect because in lines 43-44 you state that “These fish growth and condition metrics will be modelled as a function of mean chl-a”, and the mean chl-a value will be site-specific, rather than at an atoll scale.

Line 41: The text states that a linear mixed effects model will be fitted to the response variable “fish growth rate”. However, page 7 line 5 says otolith microstructure will be examined to determine “fish growth rate”, and page 7 lines 33-35 that “parameters of growth” will be estimated using the growth function equation. For clarity, specify which growth rate parameter is being tested in the LME, and how it was derived.

Lines 44-48: “we will examine for an interaction between a) depth of collection and mean chl-a... for each response variable” – how do you intend to examine this interaction if both depths would be assigned the same mean chl-a value per site? You cannot extrapolate satellite-derived chl-a values across depth?

Lines 44-48: “we will examine for an interaction between ... b) depth of collection and site slope ... for each response variable” – will a different reef slope steepness value (angle?) be calculated for shallow versus moderate depths for each site? Or will each site have one reef slope steepness value? Are depth transponder data binned in some way, or continuous? Please clarify.

Overall, it was confusing that lines 44-48 state that the parameters will be examined for an interaction between a) depth of collection and mean chl-a, and also an interaction between b) depth of collection and site slope, but then the design table (pages 10-13) presents H1-H12 that examine individual factors for each parameter.

Page 12:

Please clarify the wording in hypotheses H7 and H8 in relation to the prediction stated at the end of the Introduction (as per below). The two seem to be contradicting, or otherwise the wording needs to be more clear – is it hypothesized that d15N values will be enriched at 1) shallow sites with 2) gradual slopes? It is unclear whether the wording “shallower reef slope” pertains to sampling depth (testing shallow vs. moderate depths) and/or reef slope (depth transponder data)?

Hypothesis H7: “Fish nitrogen isotope d15N values will be enriched at sites with shallower reef slopes facilitating physical nutrient delivery mechanisms” and H8 “fish d13C will be depleted at sites with shallower reef slopes...”

Page 4, lines 58-60: “we predict that fish collected at greater depths will show enriched d15N and depleted d13C values indicating an increased reliance on primary production derived from deeper oceanic nutrient sources”

The wording “shallower reef slope” also needs to be clarified in H5 and H6.

Review form: Reviewer 2

Do you have any ethical concerns with this paper?

No

Recommendation?

Accept with minor revision

Comments to the Author(s)

This looks like an interesting study with a well thought out design. I just have a few comments about proposed analytical approaches and methodological clarity.

1) I'm a bit concerned about the mixed model set up, with a 3-level factor for island as a random intercept. REs with fewer than 5 levels aren't ideal - at best you get wonky model estimates or at worst non-converging models. It's also not made clear how you would deal with the site-level variation. I'd be more comfortable with a nested island/site random intercept, and indeed this 'variation among site within islands' quantity might be quite interesting for your analysis.

2) Related to the above, it's not really clear how you will conduct the permutation tests. You report per-depth sample sizes which look like they are pooling all site-level data (which indeed you state on page 9. But samples from the different sites won't be exchangeable, so ideally you'd constrain the permutations so that whole blocks of sites are randomly re-assigned, rather than individuals. This likely means your power won't be as high as indicated, unless you've controlled for these constraints in your analysis already.

3) You're testing around 24 hypotheses, and it's great that you enumerate these to acknowledge the extent of testing. But you list an alpha of 0.05 with no indication that you'll be correcting for multiple testing. Have I missed this in the text?

Page 9: I don't think 'permutation-testing' needs a hyphen

Page 10: For H1, you mention a total sample size of 18 - but I assume this refers to sites not sum of individuals across sites?

Review form: Reviewer 3

Do you have any ethical concerns with this paper?

No

Recommendation?

Reject

Comments to the Author(s)

1. General comments

This study proposes to explore fish growth inferred from otolith chemistry in the context of oceanic productivity. Oceanic controls on reef processes is a very interesting topic but it is unclear why these sites would be expected to differ in their oceanographic conditions and thus the provision of oceanic nutrients to reef slope communities. Since the proposal is to only assess chlorophyll-a concentrations, i.e. the standing stock of phytoplankton, by satellite, without any

kind of ground-truthing, the study does not seem likely to adequately assess oceanic productivity itself. An especially noticeable aspect overlooked in such an approach is the variation in the standing stock and productivity of zooplankton, which is the most likely the vector to link these reef fish with the ocean. The proposed stable isotope approach is also highly rudimentary and poses great risk of misinterpretation – variation, or lack thereof, in bulk stable isotopes are a function of a great many factors that are not assessed, or even justified, in the proposed approach. In addition to food sources, bulk stable isotope variation can reflect changes in isotopic baselines and variations in discrimination related to physiology and metabolism (e.g. growth rates). There is no essential sampling of isotope baselines in space or time at these sites, or any exploration of discrimination variation likely across environmental gradients, and the proposed analysis has the strong potential to represent another ambiguous coral reef isotope data set with limited robust inference potential.

2. Detailed comments

2.1. Major comments

1. The text suffers from extremely limited referencing. In many cases only very recent papers (post 2010, rarely before 2000) are cited. This leads to some contentious or incorrect statements about the biogeochemistry of reef ecosystems and stable isotope analyses. A revision of some earlier seminal works, as well as key concepts on reef biogeochemistry, would strengthen or correct many of the statements in the text. Example references are provided below for some, but not all, of the contentious statements.

2. The quantification of oceanic productivity, and the capacity to demonstrate differences between sites, seems poorly justified and likely to mislead. The analysis relies wholly on the standing stock of phytoplankton (chl-a concentrations) in the very surface layer of the ocean from satellite. There is no ground-truthing with in situ sampling. The relationships between primary and higher productivity are ambiguous. Perhaps most importance, despite the length text exploring the role of vertical structure on nutrient supply to reefs, there is no exploration of the 3D structure in chl-a material potentially being supplied to different reef depths and how this could be inferred from satellites.

3. The isotope approach is inadequate for the question at hand and has limited robust inference potential. There is no quantification of isotopic baseline variation or changes in discrimination factors that will lead to variation in tissue isotopes (even if sampling were not across strong environmental gradients). The misuse of isotope analyses and reporting of over-interpreted isotope data is a growing problem in the literature and needs to be strongly guarded against.

4. Please number lines sequentially so that the reviewer can reference comments easily (any journal instructions to the contrary are non-sensical; here the line numbering does not match the text lines). The below comments are referenced to the line numbers provided; pages will have to be discerned by the authors.

2.2. Minor comments

5. L35-36: in addition to sources, different $\delta^{15}\text{N}$ and $\delta^{13}\text{C}$ tissue values could also represent changes in discrimination due to e.g. physiology or metabolism, which is especially likely across a large depth gradient and thus temperature conditions. Given the purpose of the paper is to interpret differences in growth, the authors are encouraged to explore the literature on growth effects on tissue stable isotopes.

6. L9-10: The statement “This productivity paradox has predominantly been explained by internal reef processes” is highly contentious and seems to arise from extremely limited referencing (a problem throughout). There are abundant studies demonstrating that, despite low nutrient concentrations, the uptake of nutrients from the ocean are not low, and the paradox thus perhaps nonexistent. Suggest revision of this statement and others, and where appropriate referencing, of at least some of these studies/texts on reef biogeochemical function: (Wiebe et al., 1975; Andrews & Gentien, 1982; Atkinson, 1992; Atkinson & Bilger, 1992; Atkinson &

Falter, 2003; Lesser et al., 2007; Genin et al., 2009; Wyatt et al., 2010; Patten et al., 2011; Wyatt et al., 2012a).

7. L14-16: The concept that DOM recycling by sponges is a major driver of reef biogeochemistry is very hard to support (albeit this paper made its way into Science; by the way the first author's surname is de Goeij, suggest correcting the reference). A much more holistic treatment of the literature is needed here, including more expansive reading and referencing on the host of mechanisms influential in reef nutrient cycling in addition to DOM cycling by sponges.

8. L19-20: Again, the referencing is limited, and only includes very recent papers. Try also: (Hamner et al., 1988; Hamner et al., 2007; Wyatt et al., 2010, 2013; Hanson et al., 2016). The focus on external nutrient subsidies stimulating "planktonic" food chains seems to miss the documented input of such oceanic subsidies into benthic food chains (this could be a wording issue, so suggest revising to make clearer).

9. L23: Here, is primary productivity referring to pelagic or benthic, or both? Suggest to clear this wording and concept up.

10. L33-37: Oceanic inputs do not only occur from upwelling of deep water dissolved nutrients. Refer to the above studies to make this statement more accurate and better referenced.

11. L38-38: It is unclear why this focuses exclusively on internal waves, and devotes a large amount of text space to them, when these are not being assessed in the study.

12. L3-5: Season and geographic location also strongly determine internal-wave interactions with reefs (Leichter et al., 2012; Wyatt et al., 2020).

13. L10-14: Again, very poorly referenced, at a minimum the ideas about heterotrophy and coral resilience have to include reference to robust studies like Grottoli et al. (2006).

14. L14-19: Please provide detailed referencing and analysis of the statement of correlation between chl-a (standing stock) and productivity based on reference to robust studies from the oceanographic literature, preferably with region specific analyses. Start with Steele and Baird (1961).

15. L17-19: Similar, strong justification is needed for the statement "satellite derived chlorophyll-a (chl-a) estimates, which correlate strongly with primary production throughout the photic zone" – how can a surface estimate of chl-a abundance say anything about subsurface abundances, let alone subsurface productivity. This becomes even more serious issue in the context of the 3D aspects of reef hydrodynamics and nutrient delivery introduced in the earlier text; satellite data is likely to be severely limited in its capacity to accurately characterize reef conditions except in the very near surface (a few m or less) (Leichter et al., 2006; Cyronak et al., 2020; Wyatt et al., 2020).

16. L19-21: Must reference at least Hanson et al. (2016).

17. L31-32: Stable isotope analysis can, at best, be used to INFER nutrient sources. This limitation is especially pronounced for bulk isotope analyses and un-robust analyses that are not constrained by experimental validation of discrimination factors and isotope baseline variation. The misuse of isotope analyses and reporting of over-interpreted isotope data is a growing problem in the literature and needs to be strongly guarded against.

18. L34: Why is only one reference published in the last year provided for stable isotope studies of reef fish nutrients sources? Please review the literature more widely and reference more appropriately.

19. L36-38: The text "distinctive basal isotopic signatures dependent on the contribution of differing primary production sources" is very awkward and difficult to understand.

20. L38-43: Referencing needs improving, at least include reference to ideas from: (Wyatt et al., 2012b; McMahon et al., 2016; Morais & Bellwood, 2019; Skinner et al., 2019).

21. L53: Please include references for C:N being a good proxy for reef fish condition. Consider robustly defining what is meant by condition.

22. L12-23: Why these sites? What is the a-priori information that would lead to a hypothesize of any difference in oceanic nutrient supply between these sites?

23. L46: A depth relationship will be next to impossible to robustly determine from sampling at two depths.
24. L30-34: There is standard, accepted best practice in reporting isotope analyses that should be adhered to, including terminology (Coplen, 2011; Skrzypek, 2013; Coleman & Meier-Augenstein, 2014). In particular, this text lacks information on the normalization procedure and the identity of standard materials.
25. L20-50: Bulk tissue stable isotope data CANNOT be interpreted in the absence of detailed quantification of the isotope baselines in space and time and understanding of the role of physiology and metabolism leading to isotopic variation. The proposed analysis has the strong potential to represent another ambiguous coral reef isotope data set with limited robust inference potential.
26. L44-51: Arithmetic lipid normalisation needs better justification, including much wider and up to date referencing (e.g. Skinner et al., 2016). Why apply lipid normalisation rather than lipid removal if there truly is a lipid concern? What was the lipid content of the samples (not the C:N)? What was the verified isotope effect of lipid removal for these samples?
27. L51-on: Satellite inferences would seem to need to be ground-truthed to be quantitatively meaningful in this context.

2.3. References

- Andrews, J.C. and Gentien, P. (1982) Upwelling as a source of nutrients to the Great Barrier Reef ecosystem: A solution to Darwin's question? *Marine Ecology Progress Series* 8: 257-269.
- Atkinson, M.J. (1992) Productivity of the Enewetak flats predicted from mass transfer relationships. *Continental Shelf Research* 12: 799-807.
- Atkinson, M.J. and Bilger, R.W. (1992) Effects of water velocity on phosphate uptake in coral reef flat communities. *Limnology and Oceanography* 37: 273-279.
- Atkinson, M.J. and Falter, J.L. (2003) Coral reefs. In: Black, K., Shimmield, G. (eds) *Biogeochemistry of marine systems*. CRC Press, Boca Raton, Florida, pp 40-64.
- Coleman, M. and Meier-Augenstein, W. (2014) Ignoring IUPAC guidelines for measurement and reporting of stable isotope abundance values affects us all. *Rapid Communications in Mass Spectrometry* 28: 1953-1955.
- Coplen, T.B. (2011) Guidelines and recommended terms for expression of stable-isotope-ratio and gas-ratio measurement results. *Rapid Communications in Mass Spectrometry* 25: 2538-2560.
- Cyronak, T., Takeshita, Y., Courtney, T.A., DeCarlo, E.H., Eyre, B.D., Kline, D.I., Martz, T., Page, H., Price, N.N., Smith, J., Stoltenberg, L., Tresguerres, M. and Andersson, A.J. (2020) Diel temperature and pH variability scale with depth across diverse coral reef habitats. *Limnology and Oceanography Letters* 5: 193-203.
- Genin, A., Monismith, S.G., Reidenbach, M.A., Yahel, G. and Koseff, J.R. (2009) Intense benthic grazing of phytoplankton in a coral reef. *Limnology and Oceanography* 54: 938-951.
- Grottoli, A.G., Rodrigues, L.J. and Palardy, J.E. (2006) Heterotrophic plasticity and resilience in bleached corals. *Nature* 440: 1186-1189.
- Hamner, W.M., Colin, P.L. and Hamner, P.P. (2007) Export-import dynamics of zooplankton on a coral reef in Palau. *Marine Ecology-Progress Series* 334: 83-92.
- Hamner, W.M., Jones, M.S., Carleton, J.H., Hauri, I.R. and Williams, D.M. (1988) Zooplankton, planktivorous fish, and water currents on a windward reef face: Great Barrier Reef, Australia. *Bulletin of Marine Science* 42: 459-479.
- Hanson, K.M., Schnarr, E.L. and Leichter, J.J. (2016) Non-random feeding enhances the contribution of oceanic zooplankton to the diet of the planktivorous coral reef fish *Dascyllus flavicaudus*. *Marine Biology* 163.
- Leichter, J.J., Helmuth, B. and Fischer, A.M. (2006) Variation beneath the surface: Quantifying complex thermal environments on coral reefs in the Caribbean, Bahamas and Florida. *Journal of Marine Research* 64: 563-588.

- Leichter, J.J., Stokes, M.D., Hench, J.L., Witting, J. and Washburn, L. (2012) The island-scale internal wave climate of Moorea, French Polynesia. *Journal of Geophysical Research-Oceans* 117: C06008.
- Lesser, M.P., Falcon, L.I., Rodriguez-Roman, A., Enriquez, S., Hoegh-Guldberg, O. and Iglesias-Prieto, R. (2007) Nitrogen fixation by symbiotic cyanobacteria provides a source of nitrogen for the scleractinian coral *Montastraea cavernosa*. *Marine Ecology-Progress Series* 346: 143-152.
- McMahon, K.W., Thorrold, S.R., Houghton, L.A. and Berumen, M.L. (2016) Tracing carbon flow through coral reef food webs using a compound-specific stable isotope approach. *Oecologia* 180: 809-821.
- Morais, R.A. and Bellwood, D.R. (2019) Pelagic Subsidies Underpin Fish Productivity on a Degraded Coral Reef. *Current Biology* 29: 1521-1527 e1526.
- Patten, N.L., Wyatt, A.S.J., Lowe, R.L. and Waite, A.M. (2011) Uptake of picophytoplankton, bacterioplankton and virioplankton by a fringing coral reef community (Ningaloo Reef, Australia). *Coral Reefs* 30: 555-567.
- Skinner, C., Newman, S.P., Mill, A.C., Newton, J. and Polunin, N.V.C. (2019) Prevalence of pelagic dependence among coral reef predators across an atoll seascape. *Journal of Animal Ecology* 88: 1564-1574.
- Skinner, M.M., Martin, A.A. and Moore, B.C. (2016) Is lipid correction necessary in the stable isotope analysis of fish tissues? *Rapid Communications in Mass Spectrometry* 30: 881-889.
- Skrzypek, G. (2013) Normalization procedures and reference material selection in stable HC/NOS isotope analyses: an overview. *Analytical and Bioanalytical Chemistry* 405: 2815-2823.
- Steele, J.H. and Baird, I.E. (1961) Relations between primary production, chlorophyll and particulate carbon. *Limnology and Oceanography* 6: 68-78.
- Wiebe, W.J., Johannes, R.E. and Webb, K.L. (1975) Nitrogen fixation in a coral reef community. *Science* 188: 257-259.
- Wyatt, A.S.J., Falter, J.L., Lowe, R.J., Humphries, S. and Waite, A.M. (2012a) Oceanographic forcing of nutrient uptake and release over a fringing coral reef. *Limnology and Oceanography* 57: 401-419.
- Wyatt, A.S.J., Leichter, J.J., Toth, L.T., Miyajima, T., Aronson, R.B. and Nagata, T. (2020) Heat accumulation on coral reefs mitigated by internal waves. *Nature Geoscience* 13: 28-34.
- Wyatt, A.S.J., Lowe, R.J., Humphries, S. and Waite, A.M. (2010) Particulate nutrient fluxes over a fringing coral reef: relevant scales of phytoplankton production and mechanisms of supply. *Marine Ecology-Progress Series* 405: 113-130.
- Wyatt, A.S.J., Lowe, R.J., Humphries, S. and Waite, A.M. (2013) Particulate nutrient fluxes over a fringing coral reef: Source-sink dynamics inferred from carbon to nitrogen ratios and stable isotopes. *Limnology and Oceanography* 58: 409-427.
- Wyatt, A.S.J., Waite, A.M. and Humphries, S. (2012b) Stable isotope analysis reveals community-level variation in fish trophodynamics across a fringing coral reef. *Coral Reefs* 31: 1029-1044.

Review form: Reviewer 4

Do you have any ethical concerns with this paper?

No

Recommendation?

Accept with minor revision

Comments to the Author(s)

The authors propose an investigation of the roles in proximity to deep water nutrients, primary productivity and depth influence the growth and condition of coral reef fishes.

The proposed methods of investigation include; a) fish morphometrics (body length, number of annuali in otoliths), b) elemental chemistry (relative ratios of total and isotopic weights of C and N) and, c) satellite derived oceanographic data.

Overall, I think it is an interesting and worthy investigation that will contribute toward understanding the effects depth on reef fish ecology - an area of growing interest that has a substantial lack of empirical ecological knowledge.

There are a couple of areas within the proposed methods that I think would make for a clearer and more compelling investigation.

The most important aspects of these are that: 1) I believe there should be greater clarity in the presentation of statistical modelling methods. 2) I have also identified a number of potential confounding factor that could mask positive causal relationships, if not accounted for. Some are likely more critical than others. However, in saying that all but one can be implemented at the data analysis or pre-analysis phases.

Statistical Models:

Pg 9, ln 38: It is generally considered that the variance within random effects is rarely modeled accurately for factors with fewer than 5 - 6 levels. Given that there are only 3 atolls included in this investigation, I would suggest that the authors will have a clearer understanding of their data if they use log likelihood tests to compare models with and without atoll as a fixed factor. Based on the test results, the atoll factor can be formally incorporated in, or dropped from, final models.

In the same context, it would be appropriate and desirable to include site as a random level factor in all models (5-6 total sites at each atoll). If the atoll that fish were collected at significantly affects the data (tested as above), then sites should be nested within atolls.

Pg 9, ln 46: The construction of the proposed full model is a little unclear in this description. Is there a three-way interaction (meanChl-a*Slope*Depth)? If not, what is the reasoning behind the assumption that an interaction between slope and mean Chl-a would not have possible effects on fish growth rate, fish condition, and isotopic value signatures in fish tissues?

Pg 9, ln 52: It seems that using a log likelihood approach comparing the full models proposed in the above paragraph (and atolls as mentioned above) to a depth only model would identify whether or not it is appropriate to pool data from all sites into a depth analysis. I am willing to hear further rationale for taking this pooling approach a-priori.

Pg 10, Hypotheses table:

As a note, there must be a clearer way to communicate the main hypotheses. There are 32 listed here.

It looks as though there is sufficient power to address the proposed questions.

I did note, however, that there is not a (clearly labelled) breakdown of the number of samples by depth and site for hypotheses that include depth comparisons. If 7 is the total number of samples across both depths at some sites, the inclusion of these sites may not aid in the constraint of data variance among depths.

Potential confounding factors that could be tested:

These are some avenues that the investigators might want to explore to better understand their data. This may be particularly important if causal relationships are not found in their primary response variables. It is likely that some of these tests, if adopted, would be best presented in supplemental materials, so as not to detract from the results main investigation.

1) Is there a relationship between variability in the delivery of deep oceanic nutrients and primary productivity? - correlation analyses? formal model interactions, as above?

2) Could nutrient delivery via terrestrial/ lagoonal sources be a confounding factor?

The investigators clearly state their sampling was designed to mitigate this effect. I would suggest initially using distance from lagoon entrance/river mouths (there are islands, right?) as a co-variate to check that this potential effect was mitigated. Again, log likelihood tests of models with and without this factor would ascertain any effect.

3) Exposure: Productivity, surge and wave action (water mixing) all vary with reef exposure. I would suggest formally investigating the potential role of exposure in all models. This could be achieved with three exposure levels (windward, leeward, oblique), or with the cardinal directions if exposure is not known.

4) Fish size: fish size can, in some cases, alter C:N ratios (e.g. Sterner and George 2000, Ecology). I suggest regressing FL against C and N metrics to ascertain that this is not a further source of variance in the data for this investigation.

5) C:N - lipid ratios: Fagan et al (2011 - Canadian Journal of fish & aquatic science) suggest that multi-taxa, multi-tissue models of C:N - Lipid ratios, such as presented by Post et al. 2007 - and cited in this proposal) may not always hold. Fagan et al. therefore suggest that species level relationships should first be investigated, where possible. If possible to do so, this may give the investigators greater confidence in the conclusions presented by their data.

Further points for clarification:

page 5, ln 46: Please clarify the actual number of samples collected by atoll, site, and depth (via a simple table).

Page 8, ln 43: What level of spatial accuracy will be achieved in overlaying satellite grids of this resolution on the sites? I am sure the authors have ascertained that the resolution is appropriate for the study, but it would be amiss not to check - given that units are not given for the pixel size.

Decision letter (RSOS-201012.R0)

Dear Dr Roche,

The Editors assigned to your Stage 1 Registered Report ("Linking variation in planktonic primary production to coral reef fish growth and condition") have now received comments from reviewers. We would like you to revise your paper in accordance with the referee and editors suggestions which can be found below (not including confidential reports to the Editor). Please note this decision does not guarantee eventual acceptance.

Please submit a copy of your revised paper within three months (i.e. by the 13-Aug-2020). Please note that the revision deadline will expire at 00.00am on this date. If we do not hear from you within this time then it will be assumed that the paper has been withdrawn. In exceptional circumstances, extensions may be possible if agreed with the Editorial Office in advance.

on behalf of Professor Chris Chambers (Registered Reports Editor, Royal Society Open Science)
openscience@royalsociety.org

Associate Editor Comments to Author (Professor Chris Chambers):

Associate Editor: 1

Comments to the Author:

Four expert reviewers have now provided extremely helpful and detailed assessments of the Stage 1 manuscript. As you will see, Reviewers 1, 2 and 4 are broadly positive, noting the importance and timeliness of the research question, while also offering a wide range constructive suggestions from improvement, including consideration of additional measurements and positive controls, greater methodological detail, and clarification of the statistical analyses. Reviewer 3, however, is more critical and recommends rejection. Among the reviewer's major concerns is the fundamental validity of the methodology for quantifying oceanic productivity and the inferential limitations of the isotopic analyses. Indeed, the reviewer goes as far as to suggest that the use of these analyses is a serious problem in the literature.

Given the positive appraisals of the other three reviews, as well as the unique opportunity offered by the Registered Report format to improve and strengthen a study design before it is implemented, I would like to offer the authors the chance to submit a major revision. It is vital that this revision comprehensively addresses the concerns of all the reviews, and Reviewer 3 particularly, through major adjustment to the design or a very compelling rebuttal.

Concerning the design table, Reviewer 4 questions its readability in its current form. Please do keep the table in the manuscript, but I suggest presenting it in landscape format, and with a

reduced font size to improve clarity. In addition, please revise the "Interpretation given different outcomes" column to ensure that it shows which results will support or not support each hypothesis. An example of how to do this may be found in the same column of a design table on p23 onwards in this accepted Stage 1 RR: <https://osf.io/4zxc6/> I suggest modelling the language in the Interpretation column on this example accordingly.

Comments to Author:

Reviewer: 1

Comments to the Author(s)

The research question addresses the important and timely question of whether/how spatial differences in primary productivity (nutritional resources) affect fish populations. There is great interest in the relationship between primary productivity and coral reef ecosystem function, including fish growth and its contribution to biomass accumulation. Similar questions have been investigated at a global scale across fish of different dietary categories (e.g., Morais and Bellwood 2018) but the scales at which primary productivity was considered presented challenges for investigating the response of fish growth. This proposal reviewed here presents a unique research question that would allow for an in-depth examination of specific growth parameters of one species of fish across three atolls (18 sites, I believe) of an oceanic atoll system. The question is approached in a logical manner, with a thorough breakdown providing details on each of the hypotheses to be tested. The design is appropriate and considers several different important environmental aspects that may have a role in nutrient delivery across the reef slope. I recommend that this Stage 1 manuscript be accepted following minor revisions.

Additional brief detail about the focal species would be welcome (e.g., reef-associated planktivore; how far off the reef does it typically feed?).

Ideally, the project would examine $\delta^{13}\text{C}$ and $\delta^{15}\text{N}$ values of expected food sources in the water column where the fish were collected. See comment below about the inclusion of zooplankton and particulate organic matter samples, as stated in the Methods of the original submission of the manuscript.

Other potential positive controls to have an idea of baseline isotope values (but of course does not account for fractionation, which is why zooplankton samples would be better) would be to consider whether the group/collaborators have access to any specimens of benthic reef organisms that have long turn-over time for tissue C and N. I am aware that there are $\delta^{15}\text{N}$ measurements of turf algae from the reef crest from Chagos (Graham et al. 2018), but perhaps the authors know of other papers or collaborators who may have measured stable isotope ratios in benthic organisms.

Additional comments:

Please use continuous line numbering throughout the document rather than page-by-page line numbers to assist the reviewers and editors. Also, the line numbers do not appear to match up to the actual lines for some reason, so the following comments related to line numbers may be slightly off.

Abstract:

Lines 26-27: "fish growth and condition will be higher" perhaps needs to be revised to something along the lines of "parameters of fish growth and condition will be higher" otherwise "fish... condition will be higher" is unclear (because condition has not yet been defined).

Lines 43/44: Spelling - "there [were] no significant relationships..."

Lines 46-50: "...no difference in d15N or d13C values in fish collected from sites with higher mean chlorophyll-a, or at greater depths across reefs, indicating a homogeneous primary production resource" does not necessarily "suggest an absence of deep-water oceanic nutrient influence". It is possible that mixing distributes the deep-water oceanic nutrient source so that differences are not detected between the depths sampled, which are not drastically different in space (17.5 m and 10 m). It is also possible that the sampling depths simply were not deep enough to detect a potential influence of deep-water oceanic nutrient source. This should be re-written more openly so as not to dismiss the other reasons why a deep-water nutrient source was not detected, and then caveats discussed in the Discussion.

Introduction:

I was surprised not to see any mention of Letessier et al. 2016 (Enhanced pelagic biomass around coral atolls) in the Introduction as this research not only is focused on pelagic biomass and fish biomass but also presents data from Chagos (same location as this proposal). Given the close ties and relevancy to the objectives of this paper, it'd be ideal to tie this work into your Introduction.

Samoilys et al. 2018 (Patterns in reef fish assemblages: Insights from the Chagos Archipelago) presents fish biomass data (total biomass, as well as planktivore biomass) for several of the same Chagos atolls (including fore reef sites) and similar depth range that you consider in your study. This paper shows that planktivore biomass differs significantly between atolls in Chagos. Given the relevance of this, this could likely be incorporated into your Introduction as you set the scene for your question of whether fish growth rates differ across the archipelago.

It is worth mentioning in the Introduction that you will be focusing on planktivorous fish, such as in the last paragraph with the hypotheses.

Page 3, lines 9-16: The sentence "This productivity paradox...main mechanisms driving productivity" should be rewritten for clarity. It is worth to consider the wording "This productivity paradox has predominantly been explained by internal reef processes" (with the concern here being "internal reef processes") because hydrodynamic processes have a critical role in both providing nutrients and facilitating nutrient exchange.

Page 4, lines 20-21: "...in response to variability in their energy environment..." - what do you mean by "energy environment"?

Methods:

Considering the dominant southeast winds in Chagos, you may also consider the exposure of your different sites across the atolls (e.g., sheltered versus exposed to the dominant southeast winds). For example, see Perry et al. 2015 ("Remote coral reefs can sustain high growth potential and may match future sea-level trends") characterization of wave exposure across some of the same atolls that you examine in your study. Perhaps these exposure differences will be clear among your site mean chlorophyll-a values?

You likely may find that remote sensing data is quite patchy during the wet monsoon season. To this end, you may consider generating mean chlorophyll-a maps for a time period longer than only 2017-2019 if missing data affects the analysis.

Lines 12-14: When did you collect the fish? More information is needed - what were the sampling dates? The date of collection is important in relation to the environment experienced by the fish prior to the collection (as noted in the Introduction, delivery of nutrient subsidies are

affected by temporal variability). Also, the reader will want to know whether the fish were all collected in one time period, as this is important to the interpretation of the stable isotope data.

Lines 31-38: The depth recordings used to obtain information about the physical steepness of the reef slope is very interesting, and this method could be of interest to the community as reef slope parameters are not recorded often enough. Please provide further details about the method used.

Proposed Analyses:

In the original submission (RSOS-200319), page 6 lines 47-58 and page 7 lines 29-34, stated that both particulate organic matter and zooplankton were collected in the field and that you planned to complete stable isotope analyses for both of these sources. This is important in situ data in relation to the satellite-derived data. Why was this removed from the revised submission? This would be very useful data if available.

Perhaps the authors may consider standardizing K relative to the maximum size of the focal species instead of L_{∞} , as argued for and demonstrated by Morais and Bellwood 2018 (Global drivers of reef fish growth). At the least, it is worth checking whether or not L_{∞} exceeds the reported maximum size of the species by 50% or more.

Lines 8-9: Typo (“...a tin capsules”).

Line 27: The variable “ R ” (e.g., R_{sample}) is not defined for the formula presented.

Lines 48-51: You may want to add the additional detail that ... “the linear relationship has been established between C:N and lipid content [for aquatic animals]”.

Proposed Analyses, Page 3:

Line 33: Specify the spatial resolution of the MODIS Aqua chlorophyll-a data (4 km?). In Figure 1 of the original submission (RSOS-200319), some sites (per atoll) appear quite close together so it is important to state the resolution of the chlorophyll-a data (as well as other remote sensing products mentioned) and add details about how the proximity of sites will be treated in relation to assigning mean chlorophyll-a values per site.

Statistical analysis, page 9:

Lines 38-39: Due to predominant southeast wind and therefore wave exposure, it would make more sense to include individual sites as a random effect instead of the atoll/island level which encompasses sites across all sides of each atoll. It also makes more sense to have site as a random effect because in lines 43-44 you state that “These fish growth and condition metrics will be modelled as a function of mean chl-a”, and the mean chl-a value will be site-specific, rather than at an atoll scale.

Line 41: The text states that a linear mixed effects model will be fitted to the response variable “fish growth rate”. However, page 7 line 5 says otolith microstructure will be examined to determine “fish growth rate”, and page 7 lines 33-35 that “parameters of growth” will be estimated using the growth function equation. For clarity, specify which growth rate parameter is being tested in the LME, and how it was derived.

Lines 44-48: “we will examine for an interaction between a) depth of collection and mean chl-a... for each response variable” – how do you intend to examine this interaction if both depths would

be assigned the same mean chl-a value per site? You cannot extrapolate satellite-derived chl-a values across depth?

Lines 44-48: “we will examine for an interaction between ... b) depth of collection and site slope ... for each response variable” – will a different reef slope steepness value (angle?) be calculated for shallow versus moderate depths for each site? Or will each site have one reef slope steepness value? Are depth transponder data binned in some way, or continuous? Please clarify.

Overall, it was confusing that lines 44-48 state that the parameters will be examined for an interaction between a) depth of collection and mean chl-a, and also an interaction between b) depth of collection and site slope, but then the design table (pages 10-13) presents H1-H12 that examine individual factors for each parameter.

Page 12:

Please clarify the wording in hypotheses H7 and H8 in relation to the prediction stated at the end of the Introduction (as per below). The two seem to be contradicting, or otherwise the wording needs to be more clear – is it hypothesized that d15N values will be enriched at 1) shallow sites with 2) gradual slopes? It is unclear whether the wording “shallower reef slope” pertains to sampling depth (testing shallow vs. moderate depths) and/or reef slope (depth transponder data)?

Hypothesis H7: “Fish nitrogen isotope d15N values will be enriched at sites with shallower reef slopes facilitating physical nutrient delivery mechanisms” and H8 “fish d13C will be depleted at sites with shallower reef slopes...”

Page 4, lines 58-60: “we predict that fish collected at greater depths will show enriched d15N and depleted d13C values indicating an increased reliance on primary production derived from deeper oceanic nutrient sources”

The wording “shallower reef slope” also needs to be clarified in H5 and H6.

Reviewer: 2

Comments to the Author(s)

This looks like an interesting study with a well thought out design. I just have a few comments about proposed analytical approaches and methodological clarity.

1) I'm a bit concerned about the mixed model set up, with a 3-level factor for island as a random intercept. REs with fewer than 5 levels aren't ideal - at best you get wonky model estimates or at worst non-converging models. It's also not made clear how you would deal with the site-level variation. I'd be more comfortable with a nested island/site random intercept, and indeed this 'variation among site within islands' quantity might be quite interesting for your analysis.

2) Related to the above, it's not really clear how you will conduct the permutation tests. You report per-depth sample sizes which look like they are pooling all site-level data (which indeed you state on page 9. But samples from the different sites won't be exchangeable, so ideally you'd constrain the permutations so that whole blocks of sites are randomly re-assigned, rather than individuals. This likely means your power won't be as high as indicated, unless you've controlled for these constraints in your analysis already.

3) You're testing around 24 hypotheses, and it's great that you enumerate these to acknowledge the extent of testing. But you list an alpha of 0.05 with no indication that you'll be correcting for multiple testing. Have I missed this in the text?

Page 9: I don't think 'permutation-testing' needs a hyphen

Page 10: For H1, you mention a total sample size of 18 - but I assume this refers to sites not sum of individuals across sites?

Reviewer: 3

Comments to the Author(s)

1. General comments

This study proposes to explore fish growth inferred from otolith chemistry in the context of oceanic productivity. Oceanic controls on reef processes is a very interesting topic but it is unclear why these sites would be expected to differ in their oceanographic conditions and thus the provision of oceanic nutrients to reef slope communities. Since the proposal is to only assess chlorophyll-a concentrations, i.e. the standing stock of phytoplankton, by satellite, without any kind of ground-truthing, the study does not seem likely to adequately assess oceanic productivity itself. An especially noticeable aspect overlooked in such an approach is the variation in the standing stock and productivity of zooplankton, which is the most likely the vector to link these reef fish with the ocean. The proposed stable isotope approach is also highly rudimentary and poses great risk of misinterpretation - variation, or lack thereof, in bulk stable isotopes are a function of a great many factors that are not assessed, or even justified, in the proposed approach. In addition to food sources, bulk stable isotope variation can reflect changes in isotopic baselines and variations in discrimination related to physiology and metabolism (e.g. growth rates). There is no essential sampling of isotope baselines in space or time at these sites, or any exploration of discrimination variation likely across environmental gradients, and the proposed analysis has the strong potential to represent another ambiguous coral reef isotope data set with limited robust inference potential.

2. Detailed comments

2.1. Major comments

1. The text suffers from extremely limited referencing. In many cases only very recent papers (post 2010, rarely before 2000) are cited. This leads to some contentious or incorrect statements about the biogeochemistry of reef ecosystems and stable isotope analyses. A revision of some earlier seminal works, as well as key concepts on reef biogeochemistry, would strengthen or correct many of the statements in the text. Example references are provided below for some, but not all, of the contentious statements.
2. The quantification of oceanic productivity, and the capacity to demonstrate differences between sites, seems poorly justified and likely to mislead. The analysis relies wholly on the standing stock of phytoplankton (chl-a concentrations) in the very surface layer of the ocean from satellite. There is no ground-truthing with in situ sampling. The relationships between primary and higher productivity are ambiguous. Perhaps most importance, despite the length text exploring the role of vertical structure on nutrient supply to reefs, there is no exploration of the 3D structure in chl-a material potentially being supplied to different reef depths and how this could be inferred from satellites.
3. The isotope approach is inadequate for the question at hand and has limited robust inference potential. There is no quantification of isotopic baseline variation or changes in discrimination factors that will lead to variation in tissue isotopes (even if sampling were not across strong environmental gradients). The misuse of isotope analyses and reporting of over-interpreted isotope data is a growing problem in the literature and needs to be strongly guarded against.

4. Please number lines sequentially so that the reviewer can reference comments easily (any journal instructions to the contrary are non-sensical; here the line numbering does not match the text lines). The below comments are referenced to the line numbers provided; pages will have to be discerned by the authors.

2.2. Minor comments

5. L35-36: in addition to sources, different $\delta^{15}\text{N}$ and $\delta^{13}\text{C}$ tissue values could also represent changes in discrimination due to e.g. physiology or metabolism, which is especially likely across a large depth gradient and thus temperature conditions. Given the purpose of the paper is to interpret differences in growth, the authors are encouraged to explore the literature on growth effects on tissue stable isotopes.

6. L9-10: The statement “This productivity paradox has predominantly been explained by internal reef processes” is highly contentious and seems to arise from extremely limited referencing (a problem throughout). There are abundant studies demonstrating that, despite low nutrient concentrations, the uptake of nutrients from the ocean are not low, and the paradox thus perhaps nonexistent. Suggest revision of this statement and others, and where appropriate referencing, of at least some of these studies/texts on reef biogeochemical function: (Wiebe et al., 1975; Andrews & Gentien, 1982; Atkinson, 1992; Atkinson & Bilger, 1992; Atkinson & Falter, 2003; Lesser et al., 2007; Genin et al., 2009; Wyatt et al., 2010; Patten et al., 2011; Wyatt et al., 2012a).

7. L14-16: The concept that DOM recycling by sponges is a major driver of reef biogeochemistry is very hard to support (albeit this paper made its way into Science; by the way the first author’s surname is de Goeij, suggest correcting the reference). A much more holistic treatment of the literature is needed here, including more expansive reading and referencing on the host of mechanisms influential in reef nutrient cycling in addition to DOM cycling by sponges.

8. L19-20: Again, the referencing is limited, and only includes very recent papers. Try also: (Hamner et al., 1988; Hamner et al., 2007; Wyatt et al., 2010, 2013; Hanson et al., 2016). The focus on external nutrient subsidies stimulating “planktonic” food chains seems to miss the documented input of such oceanic subsidies into benthic food chains (this could be a wording issue, so suggest revising to make clearer).

9. L23: Here, is primary productivity referring to pelagic or benthic, or both? Suggest to clear this wording and concept up.

10. L33-37: Oceanic inputs do not only occur from upwelling of deep water dissolved nutrients. Refer to the above studies to make this statement more accurate and better referenced.

11. L38-38: It is unclear why this focuses exclusively on internal waves, and devotes a large amount of text space to them, when these are not being assessed in the study.

12. L3-5: Season and geographic location also strongly determine internal-wave interactions with reefs (Leichter et al., 2012; Wyatt et al., 2020).

13. L10-14: Again, very poorly referenced, at a minimum the ideas about heterotrophy and coral resilience have to include reference to robust studies like Grottoli et al. (2006).

14. L14-19: Please provide detailed referencing and analysis of the statement of correlation between chl-a (standing stock) and productivity based on reference to robust studies from the oceanographic literature, preferably with region specific analyses. Start with Steele and Baird (1961).

15. L17-19: Similar, strong justification is needed for the statement “satellite derived chlorophyll-a (chl-a) estimates, which correlate strongly with primary production throughout the photic zone” – how can a surface estimate of chl-a abundance say anything about subsurface abundances, let alone subsurface productivity. This becomes even more serious issue in the context of the 3D aspects of reef hydrodynamics and nutrient delivery introduced in the earlier text; satellite data is likely to be severely limited in its capacity to accurately characterize reef conditions except in the very near surface (a few m or less) (Leichter et al., 2006; Cyronak et al., 2020; Wyatt et al., 2020).

16. L19-21: Must reference at least Hanson et al. (2016).

17. L31-32: Stable isotope analysis can, at best, be used to INFER nutrient sources. This limitation is especially pronounced for bulk isotope analyses and un-robust analyses that are not constrained by experimental validation of discrimination factors and isotope baseline variation. The misuse of isotope analyses and reporting of over-interpreted isotope data is a growing problem in the literature and needs to be strongly guarded against.
18. L34: Why is only one reference published in the last year provided for stable isotope studies of reef fish nutrients sources? Please review the literature more widely and reference more appropriately.
19. L36-38: The text “distinctive basal isotopic signatures dependent on the contribution of differing primary production sources” is very awkward and difficult to understand.
20. L38-43: Referencing needs improving, at least include reference to ideas from, (Wyatt et al., 2012b; McMahan et al., 2016; Morais & Bellwood, 2019; Skinner et al., 2019).
21. L53: Please include references for C:N being a good proxy for reef fish condition. Consider robustly defining what is meant by condition.
22. L12-23: Why these sites? What is the a-priori information that would lead to a hypothesize of any difference in oceanic nutrient supply between these sites?
23. L46: A depth relationship will be next to impossible to robustly determine from sampling at two depths.
24. L30-34: There is standard, accepted best practice in reporting isotope analyses that should to be adhered to, including terminology (Coplen, 2011; Skrzypek, 2013; Coleman & Meier-Augenstein, 2014). In particular, this text lacks information on the normalization procedure and the identity of standard materials.
25. L20-50: Bulk tissue stable isotope data CANNOT be interpreted in the absence of detailed quantification of the isotope baselines in space and time and understanding of the role of physiology and metabolism leading to isotopic variation. The proposed analysis has the strong potential to represent another ambiguous coral reef isotope data set with limited robust inference potential.
26. L44-51: Arithmetic lipid normalisation needs better justification, including much wider and up to date referencing (e.g. Skinner et al., 2016). Why apply lipid normalisation rather than lipid removal if there truly is a lipid concern? What was the lipid content of the samples (not the C:N)? What was the verified isotope effect of lipid removal for these samples?
27. L51-on: Satellite inferences would seem to need to be ground-truthed to be quantitatively meaningful in this context.

2.3. References

- Andrews, J.C. and Gentien, P. (1982) Upwelling as a source of nutrients to the Great Barrier Reef ecosystem: A solution to Darwin's question? *Marine Ecology Progress Series* 8: 257-269.
- Atkinson, M.J. (1992) Productivity of the Enewetak flats predicted from mass transfer relationships. *Continental Shelf Research* 12: 799-807.
- Atkinson, M.J. and Bilger, R.W. (1992) Effects of water velocity on phosphate uptake in coral reef flat communities. *Limnology and Oceanography* 37: 273-279.
- Atkinson, M.J. and Falter, J.L. (2003) Coral reefs. In: Black, K., Shimmield, G. (eds) *Biogeochemistry of marine systems*. CRC Press, Boca Raton, Florida, pp 40-64.
- Coleman, M. and Meier-Augenstein, W. (2014) Ignoring IUPAC guidelines for measurement and reporting of stable isotope abundance values affects us all. *Rapid Communications in Mass Spectrometry* 28: 1953-1955.
- Coplen, T.B. (2011) Guidelines and recommended terms for expression of stable-isotope-ratio and gas-ratio measurement results. *Rapid Communications in Mass Spectrometry* 25: 2538-2560.
- Cyronak, T., Takeshita, Y., Courtney, T.A., DeCarlo, E.H., Eyre, B.D., Kline, D.I., Martz, T., Page, H., Price, N.N., Smith, J., Stoltenberg, L., Tresguerres, M. and Andersson, A.J. (2020) Diel temperature and pH variability scale with depth across diverse coral reef habitats. *Limnology and Oceanography Letters* 5: 193-203.

- Genin, A., Monismith, S.G., Reidenbach, M.A., Yahel, G. and Koseff, J.R. (2009) Intense benthic grazing of phytoplankton in a coral reef. *Limnology and Oceanography* 54: 938-951.
- Grottoli, A.G., Rodrigues, L.J. and Palardy, J.E. (2006) Heterotrophic plasticity and resilience in bleached corals. *Nature* 440: 1186-1189.
- Hamner, W.M., Colin, P.L. and Hamner, P.P. (2007) Export-import dynamics of zooplankton on a coral reef in Palau. *Marine Ecology-Progress Series* 334: 83-92.
- Hamner, W.M., Jones, M.S., Carleton, J.H., Hauri, I.R. and Williams, D.M. (1988) Zooplankton, planktivorous fish, and water currents on a windward reef face: Great Barrier Reef, Australia. *Bulletin of Marine Science* 42: 459-479.
- Hanson, K.M., Schnarr, E.L. and Leichter, J.J. (2016) Non-random feeding enhances the contribution of oceanic zooplankton to the diet of the planktivorous coral reef fish *Dascyllus flavicaudus*. *Marine Biology* 163.
- Leichter, J.J., Helmuth, B. and Fischer, A.M. (2006) Variation beneath the surface: Quantifying complex thermal environments on coral reefs in the Caribbean, Bahamas and Florida. *Journal of Marine Research* 64: 563-588.
- Leichter, J.J., Stokes, M.D., Hench, J.L., Witting, J. and Washburn, L. (2012) The island-scale internal wave climate of Moorea, French Polynesia. *Journal of Geophysical Research-Oceans* 117: C06008.
- Lesser, M.P., Falcon, L.I., Rodriguez-Roman, A., Enriquez, S., Hoegh-Guldberg, O. and Iglesias-Prieto, R. (2007) Nitrogen fixation by symbiotic cyanobacteria provides a source of nitrogen for the scleractinian coral *Montastraea cavernosa*. *Marine Ecology-Progress Series* 346: 143-152.
- McMahon, K.W., Thorrold, S.R., Houghton, L.A. and Berumen, M.L. (2016) Tracing carbon flow through coral reef food webs using a compound-specific stable isotope approach. *Oecologia* 180: 809-821.
- Morais, R.A. and Bellwood, D.R. (2019) Pelagic Subsidies Underpin Fish Productivity on a Degraded Coral Reef. *Current Biology* 29: 1521-1527 e1526.
- Patten, N.L., Wyatt, A.S.J., Lowe, R.L. and Waite, A.M. (2011) Uptake of picophytoplankton, bacterioplankton and virioplankton by a fringing coral reef community (Ningaloo Reef, Australia). *Coral Reefs* 30: 555-567.
- Skinner, C., Newman, S.P., Mill, A.C., Newton, J. and Polunin, N.V.C. (2019) Prevalence of pelagic dependence among coral reef predators across an atoll seascape. *Journal of Animal Ecology* 88: 1564-1574.
- Skinner, M.M., Martin, A.A. and Moore, B.C. (2016) Is lipid correction necessary in the stable isotope analysis of fish tissues? *Rapid Communications in Mass Spectrometry* 30: 881-889.
- Skrzypek, G. (2013) Normalization procedures and reference material selection in stable HC/NOS isotope analyses: an overview. *Analytical and Bioanalytical Chemistry* 405: 2815-2823.
- Steele, J.H. and Baird, I.E. (1961) Relations between primary production, chlorophyll and particulate carbon. *Limnology and Oceanography* 6: 68-78.
- Wiebe, W.J., Johannes, R.E. and Webb, K.L. (1975) Nitrogen fixation in a coral reef community. *Science* 188: 257-259.
- Wyatt, A.S.J., Falter, J.L., Lowe, R.J., Humphries, S. and Waite, A.M. (2012a) Oceanographic forcing of nutrient uptake and release over a fringing coral reef. *Limnology and Oceanography* 57: 401-419.
- Wyatt, A.S.J., Leichter, J.J., Toth, L.T., Miyajima, T., Aronson, R.B. and Nagata, T. (2020) Heat accumulation on coral reefs mitigated by internal waves. *Nature Geoscience* 13: 28-34.
- Wyatt, A.S.J., Lowe, R.J., Humphries, S. and Waite, A.M. (2010) Particulate nutrient fluxes over a fringing coral reef: relevant scales of phytoplankton production and mechanisms of supply. *Marine Ecology-Progress Series* 405: 113-130.
- Wyatt, A.S.J., Lowe, R.J., Humphries, S. and Waite, A.M. (2013) Particulate nutrient fluxes over a fringing coral reef: Source-sink dynamics inferred from carbon to nitrogen ratios and stable isotopes. *Limnology and Oceanography* 58: 409-427.
- Wyatt, A.S.J., Waite, A.M. and Humphries, S. (2012b) Stable isotope analysis reveals community-level variation in fish trophodynamics across a fringing coral reef. *Coral Reefs* 31: 1029-1044.

Reviewer: 4

Comments to the Author(s)

The authors propose an investigation of the roles in proximity to deep water nutrients, primary productivity and depth influence the growth and condition of coral reef fishes.

The proposed methods of investigation include; a) fish morphometrics (body length, number of annuali in otoliths), b) elemental chemistry (relative ratios of total and isotopic weights of C and N) and, c) satellite derived oceanographic data.

Overall, I think it is an interesting and worthy investigation that will contribute toward understanding the effects depth on reef fish ecology - an area of growing interest that has a substantial lack of empirical ecological knowledge.

There are a couple of areas within the proposed methods that I think would make for a clearer and more compelling investigation.

The most important aspects of these are that: 1) I believe there should be greater clarity in the presentation of statistical modelling methods. 2) I have also identified a number of potential confounding factor that could mask positive causal relationships, if not accounted for. Some are likely more critical than others. However, in saying that all but one can be implemented at the data analysis or pre-analysis phases.

Statistical Models:

Pg 9, ln 38: It is generally considered that the variance within random effects is rarely modeled accurately for factors with fewer than 5 - 6 levels. Given that there are only 3 atolls included in this investigation, I would suggest that the authors will have a clearer understanding of their data if they use log likelihood tests to compare models with and without atoll as a fixed factor. Based on the test results, the atoll factor can be formally incorporated in, or dropped from, final models.

In the same context, it would be appropriate and desirable to include site as a random level factor in all models (5-6 total sites at each atoll). If the atoll that fish were collected at significantly affects the data (tested as above), then sites should be nested within atolls.

Pg 9, ln 46: The construction of the proposed full model is a little unclear in this description. Is there a three-way interaction (meanChl-a*Slope*Depth)?

If not, what is the reasoning behind the assumption that an interaction between slope and mean Chl-a would not have possible effects on fish growth rate, fish condition, and isotopic value signatures in fish tissues?

Pg 9, ln 52: It seems that using a log likelihood approach comparing the full models proposed in the above paragraph (and atolls as mentioned above) to a depth only model would identify whether or not it is appropriate to pool data from all sites into a depth analysis. I am willing to hear further rationale for taking this pooling approach a-priori.

Pg 10, Hypotheses table:

As a note, there must be a clearer way to communicate the main hypotheses. There are 32 listed here.

It looks as though there is sufficient power to address the proposed questions.

I did note, however, that there is not a (clearly labelled) breakdown of the number of samples by depth and site for hypotheses that include depth comparisons. If 7 is the total number of samples across both depths at some sites, the inclusion of these sites may not aid in the constraint of data variance among depths.

Potential confounding factors that could be tested:

These are some avenues that the investigators might want to explore to better understand their data. This may be particularly important if causal relationships are not found in their primary response variables. It is likely that some of these tests, if adopted, would be best presented in supplemental materials, so as not to detract from the results main investigation.

1) Is there a relationship between variability in the delivery of deep oceanic nutrients and primary productivity? - correlation analyses? formal model interactions, as above?

2) Could nutrient delivery via terrestrial/ lagoonal sources be a confounding factor?

The investigators clearly state their sampling was designed to mitigate this effect. I would suggest initially using distance from lagoon entrance/river mouths (there are islands, right?) as a co-variate to check that this potential effect was mitigated. Again, log likelihood tests of models with and without this factor would ascertain any effect.

3) Exposure: Productivity, surge and wave action (water mixing) all vary with reef exposure. I would suggest formally investigating the potential role of exposure in all models. This could be achieved with three exposure levels (windward, leeward, oblique), or with the cardinal directions if exposure is not known.

4) Fish size: fish size can, in some cases, alter C:N ratios (e.g. Sterner and George 2000, Ecology). I suggest regressing FL against C and N metrics to ascertain that this is not a further source of variance in the data for this investigation.

5) C:N - lipid ratios: Fagan et al (2011 - Canadian Journal of fish & aquatic science) suggest that multi-taxa, multi-tissue models of C:N - Lipid ratios, such as presented by Post et al. 2007 - and cited in this proposal) may not always hold. Fagan et al. therefore suggest that species level relationships should first be investigated, where possible. If possible to do so, this may give the investigators greater confidence in the conclusions presented by their data.

Further points for clarification:

page 5, ln 46: Please clarify the actual number of samples collected by atoll, site, and depth (via a simple table).

Page 8, ln 43: What level of spatial accuracy will be achieved in overlaying satellite grids of this resolution on the sites? I am sure the authors have ascertained that the resolution is appropriate for the study, but it would be amiss not to check - given that units are not given for the pixel size.

Author's Response to Decision Letter for (RSOS-201012.R0)

See Appendix B.

Decision letter (RSOS-201012.R1)

This year has been very difficult for everyone, and we want to take the opportunity to thank you for your continued support in 2020.

The Royal Society Open Science editorial office will be closed from the evening of Friday 18 December 2020 until Monday 4 January 2021. We will not be responding during this time. If you have received a deadline within this time period, please contact us as soon as possible to allow us to extend the deadline. If you receive any automated messages during this time asking you to meet a deadline, we offer apologies and invite you to respond after the festive period or during normal working hours.

With our best for a peaceful festive period and New Year, and we look forward to working with you in 2021.

Dear Dr Roche

On behalf of the Editor, I am pleased to inform you that your Manuscript RSOS-201012.R1 entitled "Linking variation in planktonic primary production to coral reef fish growth and condition" has been accepted in principle for publication in Royal Society Open Science.

You may now progress to Stage 2 and complete the study as approved. Before commencing data collection we ask that you:

- 1) Update the journal office as to the anticipated completion date of your study.
- 2) Register your approved protocol on the Open Science Framework (<https://osf.io/>) or other recognised repository, either publicly or privately under embargo until submission of the Stage 2 manuscript. Please note that a time-stamped, independent registration of the protocol is mandatory under journal policy, and manuscripts that do not conform to this requirement cannot be considered at Stage 2. The protocol should be registered unchanged from its current approved state, with the time-stamp preceding implementation of the approved study design. We strongly recommend using the dedicated Registered Reports interface at <https://osf.io/rr> for registering the approved Stage 1 manuscript.

Following completion of your study, we invite you to resubmit your paper for peer review as a Stage 2 Registered Report. Please note that your manuscript can still be rejected for publication at Stage 2 if the Editors consider any of the following conditions to be met:

- The results were unable to test the authors' proposed hypotheses by failing to meet the approved outcome-neutral criteria.
- The authors altered the Introduction, rationale, or hypotheses, as approved in the Stage 1 submission.
- The authors failed to adhere closely to the registered experimental procedures. Please note that any deviations from the approved experimental procedures must be communicated to the editor immediately for approval, and prior to the completion of data collection. Failure to do so can result in revocation of in-principle acceptance and rejection at Stage 2 (see complete guidelines for further information).
- Any post-hoc (unregistered) analyses were either unjustified, insufficiently caveated, or overly dominant in shaping the authors' conclusions.
- The authors' conclusions were not justified given the data obtained.

We encourage you to read the complete guidelines for authors concerning Stage 2 submissions at <https://royalsocietypublishing.org/rsos/registered-reports#ReviewerGuideRegRep>. Please especially note the requirements for data sharing, reporting the URL of the independently

registered protocol, and that withdrawing your manuscript will result in publication of a Withdrawn Registration.

Once again, thank you for submitting your manuscript to Royal Society Open Science and we look forward to receiving your Stage 2 submission. If you have any questions at all, please do not hesitate to get in touch. We look forward to hearing from you shortly with the anticipated submission date for your stage two manuscript.

on behalf of Professor Chris Chambers (Registered Reports Editor, Royal Society Open Science)
openscience@royalsociety.org

Author's Response to Decision Letter for (RSOS-201012.R1)

See Appendix C.

Decision letter (RSOS-201012.R2)

Dear Dr Roche:

I am pleased to inform you that your manuscript entitled "Linking variation in planktonic primary production to coral reef fish growth and condition" is now accepted for publication in Royal Society Open Science.

Please remember to make any data sets or code libraries 'live' prior to publication, and update any links as needed when you receive a proof to check - for instance, from a private 'for review' URL to a publicly accessible 'for publication' URL. It is also good practice to add data sets, code and other digital materials to your reference list.

Royal Society Open Science is a fully open access journal. A payment may be due before your article is published. Our partner Copyright Clearance Center's RightsLink for Scientific Communications will contact the corresponding author about your open access options from the email domain @copyright.com (if you have any queries regarding fees, please see <https://royalsocietypublishing.org/rsos/charges> or contact authorfees@royalsociety.org).

The proof of your paper will be available for review using the Royal Society online proofing system and you will receive details of how to access this in the near future from our production office (openscience_proofs@royalsociety.org). We aim to maintain rapid times to publication after acceptance of your manuscript and we would ask you to please contact both the production office and editorial office if you are likely to be away from e-mail contact to minimise delays to

publication. If you are going to be away, please nominate a co-author (if available) to manage the proofing process, and ensure they are copied into your email to the journal.

on behalf of Professor Professor Chris Chambers (Subject Editor).

Follow Royal Society Publishing on Twitter: @RSocPublishing
Follow Royal Society Publishing on Facebook:
<https://www.facebook.com/RoyalSocietyPublishing/>
Read Royal Society Publishing's blog:
<https://royalsociety.org/blog/blogsearchpage/?category=Publishing>

Appendix A

Overall Response

We thank the Editor and associate editor for their comments and the information provided, which were extremely helpful. On the basis of that information we have now given additional clarity and detail, and have included a design table, listed hypotheses and power analyses for the research questions in this study.

Specific Responses

Editor Comments:

1. Make clear the total sample size that will be subjected to statistical analysis using linear mixed models. This was not immediately apparent from the Method.

Response: The total sample size for each statistical test to be carried out is now included in the design table.

2. Consider a prospective power analysis or justify the absence of a formal sampling plan. For linear mixed models, simulation methods may be the most appropriate approach for calculating power (eg. <https://besjournals.onlinelibrary.wiley.com/doi/pdf/10.1111/2041-210X.12504>) but the authors are free to choose an approach to their liking. For frequentist statistical methods, reviewers are likely to expect some formal consideration of power during Stage 1 RR assessment. Even though the sample size is likely to be fixed (given that the data are already collected), you could report a sensitivity power analysis which calculates the effect size that your design has reasonable power (e.g. 80% or 90%) to detect given the available sample size.

Response: Simulation methods (using the simr package for the linear mixed models, and the emon package for permutation tests) have now been used to calculate power for each hypothesis. As our sample sizes are fixed, the number of samples obtained for each hypothesis was used to generate 999 simulations, giving the minimum effect size detectable, with a power 80% or above for each test.

3. Ensure a direct and precise correspondence between the hypotheses, the critical statistical tests or test components that will test those hypotheses, if appropriate the statistical power of each test or test component, and the interpretation given different outcomes. To ensure maximum clarity please include in the Method section a design table as shown in section 9 of this template (<https://osf.io/93znh/>) outlining the research question, hypothesis, sampling plan, analysis plan, and which outcomes will lead to which interpretation. Attached to this letter is a PDF file containing some examples from previous submissions of how these tables can look (from different research fields to the current proposal, but the principles are the same for all statistical hypothesis testing)

Response: A design table has now been prepared based on both section 9 of the template and the two pdf examples given. It includes sections on Research Question, Hypothesis, Sample size, Power, Statistical Test, and Interpretation given different outcomes.

*4. The abstract includes (in yellow highlight) two potential summaries of the results. This is an admirable commitment to prespecification, however it seems possible that the results could be more complex and may eventually require a different statement altogether. Therefore, it is recommended that the authors replace "will" with "may" in the accompanying note: "Depending on the results obtained sections from the either the second or the third portion of the abstract highlighted **may** be used as appropriate."*

Response: The text has been changed as suggested.

It is acknowledged that the results could be more complex than the two alternatives given within the abstract. This was our effort to write the abstract prior to results being known, We will utilise sections from either portion of the abstract as may be appropriate once results are known.

Appendix B

Dear Dr Roche,

The Editors assigned to your Stage 1 Registered Report ("Linking variation in planktonic primary production to coral reef fish growth and condition") have now received comments from reviewers. We would like you to revise your paper in accordance with the referee and editors suggestions which can be found below (not including confidential reports to the Editor). Please note this decision does not guarantee eventual acceptance.

Please submit a copy of your revised paper within three months (i.e. by the 13-Aug-2020). Please note that the revision deadline will expire at 00.00am on this date. If we do not hear from you within this time then it will be assumed that the paper has been withdrawn. In exceptional circumstances, extensions may be possible if agreed with the Editorial Office in advance.

on behalf of Professor Chris Chambers (Registered Reports Editor, Royal Society Open Science) openscience@royalsociety.org

Associate Editor Comments to Author (Professor Chris Chambers):

Comments to the Author:

Four expert reviewers have now provided extremely helpful and detailed assessments of the Stage 1 manuscript. As you will see, Reviewers 1, 2 and 4 are broadly positive, noting the importance and timeliness of the research question, while also offering a wide range constructive suggestions from improvement, including consideration of additional measurements and positive controls, greater methodological detail, and clarification of the statistical analyses. Reviewer 3, however, is more critical and recommends rejection. Among the reviewer's major concerns is the fundamental validity of the methodology for quantifying oceanic productivity and the inferential

limitations of the isotopic analyses. Indeed, the reviewer goes as far as to suggest that the use of these analyses is a serious problem in the literature.

Given the positive appraisals of the other three reviews, as well as the unique opportunity offered by the Registered Report format to improve and strengthen a study design before it is implemented, I would like to offer the authors the chance to submit a major revision. It is vital that this revision comprehensively addresses the concerns of all the reviews, and Reviewer 3 particularly, through major adjustment to the design or a very compelling rebuttal.

Response: We have comprehensively revised this manuscript, addressing the concerns of all reviewers. In response to Reviewer 3's comments we are proposing to include particulate organic matter and zooplankton stable isotope analyses within the present submission, to address inferential limitations.

Concerning the design table, Reviewer 4 questions its readability in its current form. Please do keep the table in the manuscript, but I suggest presenting it in landscape format, and with a reduced font size to improve clarity. In addition, please revise the "Interpretation given different outcomes" column to ensure that it shows which results will support or not support each hypothesis. An example of how to do this may be found in the same column of a design table on p23 onwards in this accepted Stage 1 RR: <https://osf.io/4zrk6/> I suggest modelling the language in the Interpretation column on this example accordingly.

Response: The design table has been adjusted, and is now presented in landscape format, with a reduced font size for clarity. The "Interpretation given different outcomes" has been adjusted using similar language to the example provided.

*Comments to Author:
Reviewer: 1*

Comments to the Author(s)

The research question addresses the important and timely question of whether/how spatial differences in primary productivity (nutritional resources) affect fish populations. There is great interest in the relationship between primary productivity and coral reef ecosystem function, including fish growth and its contribution to biomass accumulation. Similar questions have been investigated at a global scale across fish of different dietary categories (e.g., Morais and Bellwood 2018) but the scales at which primary productivity was considered presented challenges for investigating the response of fish growth. This proposal reviewed here presents a unique research question that would allow for an in-depth examination of specific growth parameters of one species of fish across three atolls (18 sites, I believe) of an oceanic atoll system. The question is approached in a logical manner, with a thorough breakdown providing details on each of the hypotheses to be tested. The design is appropriate and considers several different important environmental aspects that may have a role in nutrient delivery across the reef slope. I recommend that this Stage 1 manuscript be accepted following minor revisions.

Additional brief detail about the focal species would be welcome (e.g., reef-associated planktivore; how far off the reef does it typically feed?).

Response: We have added additional relevant detail regarding the focal species, and what is known of its feeding habits (Lines 170 to 171).

Ideally, the project would examine d13C and d15N values of expected food sources in the water column where the fish were collected. See comment below about the inclusion of zooplankton and particulate organic matter samples, as stated in the Methods of the original submission of the manuscript.

Other potential positive controls to have an idea of baseline isotope values (but of course does not account for fractionation, which is why zooplankton samples would be better) would be to consider whether the group/collaborators have access to any specimens of benthic reef organisms that have long turn-over time for tissue C and N. I am aware that there are d15N measurements of turf algae from the reef crest from Chagos (Graham et al. 2018), but perhaps the authors know of other papers or collaborators who may have measured stable isotope ratios in benthic organisms.

Response: We collected zooplankton and particulate organic matter samples as part of our overall research activities during the ship based expedition. Following the feedback of reviewer 1 and reviewer 3, we are including the d15N and d13C results within this registered report, rather than incorporating them into a separate research article. See lines 188 to 191, and 233 to 242.

Additional comments:

Please use continuous line numbering throughout the document rather than page-by-page line numbers to assist the reviewers and editors. Also, the line numbers do not appear to match up to the actual lines for some reason, so the following comments related to line numbers may be slightly off.

Response: This formatting has been corrected in the revised document.

Abstract:

Lines 26-27: “fish growth and condition will be higher” perhaps needs to be revised to something along the lines of “parameters of fish growth and condition will be higher” otherwise “fish... condition will be higher” is unclear (because condition has not yet been defined).

Response: We have changed this to “parameters of fish growth and condition will be higher” as suggested. Line 18 in the revised document.

Lines 43/44: Spelling – “there [were] no significant relationships...”

Response: This has been corrected. Line 33 in revised document.

Lines 46-50: “...no difference in d15N or d13C values in fish collected from sites with higher mean chlorophyll-a, or at greater depths across reefs, indicating a homogeneous primary production resource” does not necessarily “suggest an absence of deep-water oceanic nutrient influence”. It is possible that mixing

distributes the deep-water oceanic nutrient source so that differences are not detected between the depths sampled, which are not drastically different in space (17.5 m and 10 m). It is also possible that the sampling depths simply were not deep enough to detect a potential influence of deep-water oceanic nutrient source. This should be re-written more openly so as not to dismiss the other reasons why a deep-water nutrient source was not detected, and then caveats discussed in the Discussion.

Response: We appreciate this suggestion and have revised the potential abstract text as suggested, to include alternative explanations for a lack of difference. Lines 37 to 40 in the revised document.

Introduction:

I was surprised not to see any mention of Letessier et al. 2016 (Enhanced pelagic biomass around coral atolls) in the Introduction as this research not only is focused on pelagic biomass and fish biomass but also presents data from Chagos (same location as this proposal). Given the close ties and relevancy to the objectives of this paper, it'd be ideal to tie this work into your Introduction.

Response: We have now included a reference to Letessier et al. 2016 within a relevant section of the Introduction, as a supporting piece of evidence from the same Archipelago—line 89.

Samoilys et al. 2018 (Patterns in reef fish assemblages: Insights from the Chagos Archipelago) presents fish biomass data (total biomass, as well as planktivore biomass) for several of the same Chagos atolls (including fore reef sites) and similar depth range that you consider in your study. This paper shows that planktivore biomass differs significantly between atolls in Chagos. Given the relevance of this, this could likely be incorporated into your Introduction as you set the scene for your question of whether fish growth rates differ across the archipelago.

Response: We agree with this suggestion, and have examined incorporating this reference into the Introduction, but found that it focused on the specific study site too quickly and moved away from more broadly relevant text. However, we have now incorporated Samoilys et al. 2018, and the finding of planktivore biomass differing significantly between atolls in the Chagos Archipelago, into the Methods section, and also expect to refer to it when writing the discussion.

It is worth mentioning in the Introduction that you will be focusing on planktivorous fish, such as in the last paragraph with the hypotheses.

Response: This has been corrected in line 125 of the revised document.

Page 3, lines 9-16: The sentence “This productivity paradox...main mechanisms driving productivity” should be rewritten for clarity. It is worth to consider the wording “This productivity paradox has predominantly been explained by internal reef processes” (with the concern here being “internal reef processes”) because hydrodynamic processes have a critical role in both providing nutrients and facilitating nutrient exchange.

Response: This sentence has been rewritten for clarity, and to provide a more nuanced statement as suggested. Lines 55 to 58 in revised document.

Page 4, lines 20-21: "...in response to variability in their energy environment..." – what do you mean by "energy environment"?

Response: This wording has been clarified to show we are referring to variability in primary production. Line 101 in revised document.

Methods:

Considering the dominant southeast winds in Chagos, you may also consider the exposure of your different sites across the atolls (e.g., sheltered versus exposed to the dominant southeast winds). For example, see Perry et al. 2015 ("Remote coral reefs can sustain high growth potential and may match future sea-level trends") characterization of wave exposure across some of the same atolls that you examine in your study. Perhaps these exposure differences will be clear among your site mean chlorophyll-a values?

Response: We have indeed considered the exposure of sites across atolls during the design of this study, and have selected sites at locations representing each of the main cardinal directions around atolls, to encompass a gradient of exposure. We believe that site level mean chlorophyll-a values will be influenced by, and capture important exposure differences around atolls.

You likely may find that remote sensing data is quite patchy during the wet monsoon season. To this end, you may consider generating mean chlorophyll-a maps for a time period longer than only 2017-2019 if missing data affects the analysis.

Response: We would ideally generate mean chlorophyll-a maps for longer than 2017 to 2019, but are constrained by the temporal coverage of the Copernicus satellite data, for which data are available from July 2017.

Lines 12-14: When did you collect the fish? More information is needed – what were the sampling dates? The date of collection is important in relation to the environment experienced by the fish prior to the collection (as noted in the Introduction, delivery of nutrient subsidies are affected by temporal variability). Also, the reader will want to know whether the fish were all collected in one time period, as this is important to the interpretation of the stable isotope data.

Response: We have included the exact sample date range within the manuscript. All fish were collected within one time period, rather than being spread out over multiple collection efforts over seasons or years. See line 137 in revised document.

Lines 31-38: The depth recordings used to obtain information about the physical steepness of the reef slope is very interesting, and this method could be of interest to the community as reef slope parameters are not recorded often enough. Please provide further details about the method used.

Response: We provide further details about the method used to obtain information on the physical steepness of the reef slope. See line 157 in revised document.

Proposed Analyses:

In the original submission (RSOS-200319), page 6 lines 47-58 and page 7 lines 29-34, stated that both particulate organic matter and zooplankton were collected in the field and that you planned to complete stable isotope analyses for both of these sources. This is important in situ data in relation to the satellite-derived data. Why was this removed from the revised submission? This would be very useful data if available.

Response: Particulate organic matter and zooplankton were collected in the field and we will now include the stable isotope analyses for both of these sources within the present submission. These data were removed as we were planning to include them in a separate research piece. Following the comments on Reviewer 1 and Reviewer 3 we are now including d15N and d13C results within this registered report, rather than incorporating them into a separate research article. See lines 160 to 167 and 233 to 242.

Perhaps the authors may consider standardizing K relative to the maximum size of the focal species instead of L_{∞} , as argued for and demonstrated by Morais and Bellwood 2018 (Global drivers of reef fish growth). At the least, it is worth checking whether or not L_{∞} exceeds the reported maximum size of the species by 50% or more.

Response: We thank the reviewer for this useful input. We will standardise K relative to the maximum size of *Chromis fieldi* as argued for within Morais and Bellwood 2018. Please see lines 273 to 282 in the revised document.

Lines 8-9: Typo (“...a tin capsules”).

Response: This has been corrected.

Line 27: The variable “R” (e.g., R_{sample}) is not defined for the formula presented.

Response: The variable R is now defined for the formula presented. Lines 205 to 206 in the revised document

Lines 48-51: You may want to add the additional detail that ... “the linear relationship has been established between C:N and lipid content [for aquatic animals]”.

Response: This additional detail has been added—line 221 revised document.

Proposed Analyses, Page 3:

Line 33: Specify the spatial resolution of the MODIS Aqua chlorophyll-a data (4 km?). In Figure 1 of the original submission (RSOS-200319), some sites (per atoll)

appear quite close together so it is important to state the resolution of the chlorophyll-a data (as well as other remote sensing products mentioned) and add details about how the proximity of sites will be treated in relation to assigning mean chlorophyll-a values per site.

Response: The spatial resolution of the data MODIS Level 2 data have nominal (nadir) resolution of 1 km. This is now indicated in line 295 and 296 in the revised document. No spatial averaging will be applied to the MODIS data. For validation of Sentinel 3 chl against MODIS chl (off-shore), mean, median and std.dev. will be calculated from all Sentinel 3 chl pixels that lie within a MODIS pixel. MODIS data will not be used at the actual study sites because of the proximity to land/shallow water, as noted by the reviewer. Instead, MODIS serves to validate the Sentinel 3 chl product in pixels where no land/shallow mixed pixels are present, because MODIS chl is well validated globally (e.g. Barnes et al. 2019).

Statistical analysis, page 9:

Lines 38-39: Due to predominant southeast wind and therefore wave exposure, it would make more sense to include individual sites as a random effect instead of the atoll/island level which encompasses sites across all sides of each atoll. It also makes more sense to have site as a random effect because in lines 43-44 you state that “These fish growth and condition metrics will be modelled as a function of mean chl-a”, and the mean chl-a value will be site-specific, rather than at an atoll scale.

Response: It is correct that the mean chl-a value will be site specific. We will also have two growth rate and isotope mean values per site (shallow and moderate depth), which means that we cannot include individual sites as a random effect (simply because we do not have the sample replication at the site level to do this). To clarify the sampling strategy we have added a diagram (Fig 2), and a table showing numbers of samples obtained.

Line 41: The text states that a linear mixed effects model will be fitted to the response variable “fish growth rate”. However, page 7 line 5 says otolith microstructure will be examined to determine “fish growth rate”, and page 7 lines 33-35 that “parameters of growth” will be estimated using the growth function equation. For clarity, specify which growth rate parameter is being tested in the LME, and how it was derived.

Response: The fish growth rate parameter used will be K_{max} , now indicated in line 324.

Lines 44-48: “we will examine for an interaction between a) depth of collection and mean chl-a... for each response variable” – how do you intend to examine this interaction if both depths would be assigned the same mean chl-a value per site? You cannot extrapolate satellite-derived chl-a values across depth?

Response: It is correct that we do not attempt to extrapolate satellite-derived chl-a values across depth. We examine for an interaction between depth of collection and

chl-a by comparing samples from both depths across the gradient of chl-a values we expect to see between sites.

Lines 44-48: “we will examine for an interaction between ... b) depth of collection and site slope ... for each response variable” – will a different reef slope steepness value (angle?) be calculated for shallow versus moderate depths for each site? Or will each site have one reef slope steepness value? Are depth transponder data binned in some way, or continuous? Please clarify.

Response: Each site will have one reef slope steepness value. Depth transponder data are continuous.

Overall, it was confusing that lines 44-48 state that the parameters will be examined for an interaction between a) depth of collection and mean chl-a, and also an interaction between b) depth of collection and site slope, but then the design table (pages 10-13) presents H1-H12 that examine individual factors for each parameter.

Response: This section has been edited to make it clearer that we are examining both individual factors and interactions—lines 325 to 331.

Page 12: Please clarify the wording in hypotheses H7 and H8 in relation to the prediction stated at the end of the Introduction (as per below). The two seem to be contradicting, or otherwise the wording needs to be more clear – is it hypothesized that d15N values will be enriched at 1) shallow sites with 2) gradual slopes? It is unclear whether the wording “shallower reef slope” pertains to sampling depth (testing shallow vs. moderate depths) and/or reef slope (depth transponder data)?

Response: We have changed this wording to make it clearer. We intended shallower reef slope to refer to reef slope (depth transponder data). We have now clarified to avoid potential confusion with depth by changing to “gradual reef slope” within the design table.

Hypothesis H7: “Fish nitrogen isotope d15N values will be enriched at sites with shallower reef slopes facilitating physical nutrient delivery mechanisms” and H8 “fish d13C will be depleted at sites with shallower reef slopes...”

Page 4, lines 58-60: “we predict that fish collected at greater depths will show enriched d15N and depleted d13C values indicating an increased reliance on primary production derived from deeper oceanic nutrient sources”

The wording “shallower reef slope” also needs to be clarified in H5 and H6.

Response: “shallower reef slope” has been clarified to avoid potential confusion with depth by changing to “gradual reef slope” within the design table for H5, H6, H7

Reviewer: 2

Comments to the Author(s)

This looks like an interesting study with a well thought out design. I just have a few

comments about proposed analytical approaches and methodological clarity.

1) I'm a bit concerned about the mixed model set up, with a 3-level factor for island as a random intercept. REs with fewer than 5 levels aren't ideal - at best you get wonky model estimates or at worst non-converging models. It's also not made clear how you would deal with the site-level variation. I'd be more comfortable with a nested island/site random intercept, and indeed this 'variation among site within islands' quantity might be quite interesting for your analysis.

Response: We agree that a RE with less than 5 levels is not ideal, and in response to this we now include atoll as a fixed, rather than a random effect. We clarify that as we are obtaining and analyzing site level mean values for fish isotope and growth metrics and satellite chl-a , we cannot specify site as a random effect.

2) Related to the above, it's not really clear how you will conduct the permutation tests. You report per-depth sample sizes which look like they are pooling all site-level data (which indeed you state on page 9. But samples from the different sites won't be exchangeable, so ideally you'd constrain the permutations so that whole blocks of sites are randomly re-assigned, rather than individuals. This likely means your power won't be as high as indicated, unless you've controlled for these constraints in your analysis already.

Response: We appreciate this comment, and will constrain the permutation so that blocks of sites are randomly re-assigned, rather than pooling all individual data. The power analysis has been adjusted and now reflects this.

3) You're testing around 24 hypotheses, and it's great that you enumerate these to acknowledge the extent of testing. But you list an alpha of 0.05 with no indication that you'll be correcting for multiple testing. Have I missed this in the text?

Response: We accept that there is a risk of multiple testing here. We will use the Benjamini and Hochberg (1995) procedure to control for type I errors. In implementing this procedure we will specify a false discovery rate of 0.10. We have included details on this in the statistical analysis section—lines 331 to 333 within the revised document.

Page 9: I don't think 'permutation-testing' needs a hyphen

Response: This has been corrected.

Page 10: For H1, you mention a total sample size of 18 - but I assume this refers to sites not sum of individuals across sites?

Response: This has been changed to clarify that this refers to sites, and not to the sum of individuals across sites.

Reviewer: 3

Comments to the Author(s)

1. General comments

This study proposes to explore fish growth inferred from otolith chemistry in the context of oceanic productivity. Oceanic controls on reef processes is a very interesting topic but it is unclear why these sites would be expected to differ in their oceanographic conditions and thus the provision of oceanic nutrients to reef slope communities. Since the proposal is to only assess chlorophyll-a concentrations, i.e. the standing stock of phytoplankton, by satellite, without any kind of ground-truthing, the study does not seem likely to adequately assess oceanic productivity itself. An especially noticeable aspect overlooked in such an approach is the variation in the standing stock and productivity of zooplankton, which is the most likely the vector to link these reef fish with the ocean. The proposed stable isotope approach is also highly rudimentary and poses great risk of misinterpretation – variation, or lack thereof, in bulk stable isotopes are a function of a great many factors that are not assessed, or even justified, in the proposed approach. In addition to food sources, bulk stable isotope variation can reflect changes in isotopic baselines and variations in discrimination related to physiology and metabolism (e.g. growth rates). There is no essential sampling of isotope baselines in space or time at these sites, or any exploration of discrimination variation likely across environmental gradients, and the proposed analysis has the strong potential to represent another ambiguous coral reef isotope data set with limited robust inference potential.

Response: We expect these sites to differ in their oceanographic conditions based on their positions encompassing all major cardinal directions around 3 atolls. We expect (as noted by two other reviewers) that there will be differences due to the varying exposure to wind, wave and currents which would affect the delivery of external nutrient subsidies to these reef slope sites. Based on previous studies (Baines et al., 1994; Behrenfeld & Falkowski, 1997a,b; Croll et al., 2005; Gall et al., 1999; Gove et al. 2016, Proud et al. 2017) it seems reasonable to expect that these differences will be translated into chl-a variations across sites.

We propose the inclusion of isotope results from particulate organic matter and zooplankton collected at the same sites as our fish samples to address the concerns Reviewer 3 has regarding the stable isotope approach within the study. We agree that this is an important aspect to correctly and fully interpret the results we will obtain from the fish tissue stable isotope results, and have adjusted the submission to incorporate this—see lines 160 to 167 and 233 to 242 in the revised text.

2. Detailed comments

2.1. Major comments

1. The text suffers from extremely limited referencing. In many cases only very recent papers (post 2010, rarely before 2000) are cited. This leads to some contentious or incorrect statements about the biogeochemistry of reef ecosystems and stable isotope analyses. A revision of some earlier seminal works, as well as key concepts on reef biogeochemistry, would strengthen or correct many of the statements in the text. Example references are provided below for some, but not all, of the contentious statements.

Response: We thank the reviewer for suggesting using a wider age range of references to strengthen the text. We have incorporated many of the references the reviewer has suggested, throughout the text, including some earlier seminal works.

2. The quantification of oceanic productivity, and the capacity to demonstrate differences between sites, seems poorly justified and likely to mislead. The analysis relies wholly on the standing stock of phytoplankton (chl-a concentrations) in the very surface layer of the ocean from satellite. There is no ground-truthing with in situ sampling. The relationships between primary and higher productivity are ambiguous. Perhaps most importantly, despite the length text exploring the role of vertical structure on nutrient supply to reefs, there is no exploration of the 3D structure in chl-a material potentially being supplied to different reef depths and how this could be inferred from satellites.

Response: Please see the response below to point 15 (upon which this comment is based) from within the minor comments section.

3. The isotope approach is inadequate for the question at hand and has limited robust inference potential. There is no quantification of isotopic baseline variation or changes in discrimination factors that will lead to variation in tissue isotopes (even if sampling were not across strong environmental gradients). The misuse of isotope analyses and reporting of over-interpreted isotope data is a growing problem in the literature and needs to be strongly guarded against.

Response: We propose the inclusion of isotope results from particulate organic matter and zooplankton collected at the same sites as our fish samples within this study to address the concerns Reviewer 3 has regarding the stable isotope approach within the study. We agree that this is an important aspect to correctly and fully interpret the results we will obtain from the fish tissue stable isotope results, and have adjusted the submission to incorporate this— see lines 160 to 167 and 233 to 242.

4. Please number lines sequentially so that the reviewer can reference comments easily (any journal instructions to the contrary are non-sensical; here the line numbering does not match the text lines). The below comments are referenced to the line numbers provided; pages will have to be discerned by the authors.

Response: The lines are now numbered sequentially.

2.2. Minor comments

5. L35-36: in addition to sources, different $\delta^{15}\text{N}$ and $\delta^{13}\text{C}$ tissue values could also represent changes in discrimination due to e.g. physiology or metabolism, which is especially likely across a large depth gradient and thus temperature conditions. Given the purpose of the paper is to interpret differences in growth, the authors are encouraged to explore the literature on growth effects on tissue stable isotopes.

Response: We acknowledge that $\delta^{15}\text{N}$ and $\delta^{13}\text{C}$ ratio tissue values can be due to growth effects. We have adjusted these lines in the abstract to make the language more nuanced to reflect this.

As we will obtain site level growth rate K_{max} within this study, we propose to test for the potential influences of growth rate on $\delta^{15}\text{N}$ and $\delta^{13}\text{C}$ ratio by a regression of $\delta^{15}\text{N}$ and $\delta^{13}\text{C}$ values against growth rate (K_{max}) at each site. We will place this in the supplementary material together with similar tests recommended by Reviewer 4—lines 268 to 272.

6. L9-10: *The statement “This productivity paradox has predominantly been explained by internal reef processes” is highly contentious and seems to arise from extremely limited referencing (a problem throughout). There are abundant studies demonstrating that, despite low nutrient concentrations, the uptake of nutrients from the ocean are not low, and the paradox thus perhaps nonexistent. Suggest revision of this statement and others, and where appropriate referencing, of at least some of these studies/texts on reef biogeochemical function: (Wiebe et al., 1975; Andrews & Gentien, 1982; Atkinson, 1992; Atkinson & Bilger, 1992; Atkinson & Falter, 2003; Lesser et al., 2007; Genin et al., 2009; Wyatt et al., 2010; Patten et al., 2011; Wyatt et al., 2012a).*

Response: This section has been changed to be less contentious, and to include additional referencing—lines 55 to 62.

7. L14-16: *The concept that DOM recycling by sponges is a major driver of reef biogeochemistry is very hard to support (albeit this paper made its way into Science; by the way the first author’s surname is de Goeij, suggest correcting the reference). A much more holistic treatment of the literature is needed here, including more expansive reading and referencing on the host of mechanisms influential in reef nutrient cycling in addition to DOM cycling by sponges.*

Response: We have edited this section to adopt a more balanced treatment—lines 55 to 63.

8. L19-20: *Again, the referencing is limited, and only includes very recent papers. Try also: (Hamner et al., 1988; Hamner et al., 2007; Wyatt et al., 2010, 2013; Hanson et al., 2016). The focus on external nutrient subsidies stimulating “planktonic” food chains seems to miss the documented input of such oceanic subsidies into benthic food chains (this could be a wording issue, so suggest revising to make clearer).*

Response: Many of the suggested references have been added, and these sentences have been revised to make them clearer, and to include oceanic subsidies into reef benthic food chains—lines 64 to 66.

9. L23: *Here, is primary productivity referring to pelagic or benthic, or both? Suggest to clear this wording and concept up.*

Response: This has been clarified to “pelagic primary productivity” —line 67.

10. L33-37: *Oceanic inputs do not only occur from upwelling of deep water dissolved nutrients. Refer to the above studies to make this statement more accurate and better referenced.*

Response: This sentence has been revised ” —line 73 to 74.

11. L38-38: *It is unclear why this focuses exclusively on internal waves, and devotes a large amount of text space to them, when these are not being assessed in the study.*

Response: We wish to explain and make clear to the reader why we sample fish at two different depths, and what know processes exist which could be responsible for any difference we might observe.

12. L3-5: *Season and geographic location also strongly determine internal-wave interactions with reefs (Leichter et al., 2012; Wyatt et al., 2020).*

Response: These lines have been changed to include season and geographic location and the two suggested references have been added—lines 87 to 88.

13. L10-14: *Again, very poorly referenced, at a minimum the ideas about heterotrophy and coral resilience have to include reference to robust studies like Grottoli et al. (2006).*

Response: A reference to Grottoli et al. (2006) has been added in addition to the existing reference—line 96.

14. L14-19: *Please provide detailed referencing and analysis of the statement of correlation between chl-a (standing stock) and productivity based on reference to robust studies from the oceanographic literature, preferably with region specific analyses. Start with Steele and Baird (1961).*

Response: There is a long history of modelling relationships between pelagic chl-a and primary productivity, as well as between chl-a and net primary production, beginning with Steemann Nielsen's (1952) documentation of the ^{14}C labelling technique to reliably quantify marine productivity. Whilst light, and therefore depth in the water column, places the upper limit on productivity, Talling (1957) highlighted the point that vertical motion of cells in the mixed layer means that depth-integrated production should be characterised to understand flows of carbon. The toxic, time-consuming and costly nature of ^{14}C production/productivity measurements, as well as their extremely limited spatial and temporal coverage, has further driven this effort to relate biomass (carbon or chlorophyll; Bannister, 1974; Cullen, 1990; Sathyendranath et al. 2009) to depth-integrated primary production. The superior coverage of ocean colour remote sensing data offers the possibility of estimating pelagic production roughly daily on a global scale (Perry, 1986). Early in situ measurements were based on a variety of time scales from 1–2 hours, from which a light-irradiance curve can be used to determine the maximum instantaneous rate of carbon fixation P_b^{max} , (neglects night-time respiration unless measurements are continued over the diel cycle), through to 24 hours (provides net primary production, including microbial respiration), using monospecific cultures (e.g. Ryther, 1956; Eppley & Renger, 1974)

and natural water samples (e.g. Steele & Baird, 1961; Owen & Zeitzschel, 1970). Although proportionality between chlorophyll and productivity was often shown, the proportionality constant varied in time and space (Eppley, 1972; Eppley et al., 1985; La Fontaine & Peters, 1986). The gradual accrual of datasets enabled more robust models of pelagic primary production, based on satellite estimates of chlorophyll, to be developed (e.g. Eppley et al., 1985; Platt et al., 1988; Morel, 1991; Behrenfeld et al., 2005).

Behrenfeld & Falkowski (1997b) reviewed and tested the sensitivity of the extant model types and reported that of the four key model parameters, the input biomass field and the photosynthetic efficiency (they use P_b^{opt} as appropriate for time/depth-integrated production models) drive variability.

15. L17-19: Similar, strong justification is needed for the statement “satellite derived chlorophyll-a (chl-a) estimates, which correlate strongly with primary production throughout the photic zone” – how can a surface estimate of chl-a abundance say anything about subsurface abundances, let alone subsurface productivity. This becomes even more serious issue in the context of the 3D aspects of reef hydrodynamics and nutrient delivery introduced in the earlier text; satellite data is likely to be severely limited in its capacity to accurately characterize reef conditions except in the very near surface (a few m or less) (Leichter et al., 2006; Cyronak et al., 2020; Wyatt et al., 2020).

Response: Whilst the use of satellite-derived chlorophyll-a concentration as a proxy for biomass, productivity or production is associated with uncertainties, the most intransigent being effective photosynthetic efficiency, in situ data are also associated with uncertainties. Logistics alone prevent the measurement of water-column production in situ over the life cycle of a given reef fish; many research vessels (or expensive proxy sensors) would have to be deployed continuously to gather, incubate and measure production rates across each reef. Even the measurement of sufficient in situ samples to validate remote sensing products is challenging (e.g. Campbell et al., 2002; Friedrichs et al., 2009).

We hope in future to gain funding to use extensive in situ data to validate the OLCI chl product and a representative ensemble of primary production models with which to extrapolate production estimates in space and time, using satellite data, to span the life cycle of reef fish across the Chagos Archipelago (sensu Bidigare et al., 1987; Falkowski, 1988). In the absence of such resources, there is sufficient support in the literature for the use of satellite chlorophyll as a proxy for primary production (e.g. Baines et al., 1994; Behrenfeld & Falkowski, 1997a,b; Croll et al., 2005; Gall et al., 1999; Proud et al. 2017) provided that uncertainties are considered in the analysis.

Option 1: In this analysis, we validate OLCI chl-a against MODIS (as above) and produce uncertainty maps for each in situ station.

Option 2: In this analysis, we validate OLCI chl-a against MODIS (as above) and produce chl-a uncertainty maps for each in situ station. We then apply an ensemble of primary production models to generate estimates of net primary production together with uncertainty estimates.

References:

- Baines, S.B., M.L. Pace, D.M. Karl, 1994. Why does the relationship between sinking flux and planktonic primary production differ between lakes and oceans? *Limnology & Oceanography* 39(2), 213-226.
- Bannister, T.T., 1974. Production equations in terms of chlorophyll concentration, quantum yield, and upper limit to production. *Limnology & Oceanography* 19(1), 1-12.
- Behrenfeld, M.J., P.G. Falkowski, 1997a. A consumer's guide to phytoplankton primary productivity models. *Limnology & Oceanography* 42, 1479-1491, doi:10.4319/lo.1997.42.7.1479, doi:10.4319/lo.1997.42.1.0001.
- Behrenfeld, M.J., P.G. Falkowski, 1997b. Photosynthetic rates derived from satellite-based chlorophyll concentrations. *Limnology & Oceanography* 42, 1-20.
- Behrenfeld, M.J., E. Boss, D.A. Siegel, D.M. Shea, 2005. Carbon-based ocean productivity and phytoplankton physiology from space. *Global Biogeochemical Cycles* 19, GB1006, doi:10.1029/2004GB002299.
- Bidigare, R.R., R.C. Smith, K.S. Baker, J. Marra, 1987. Oceanic primary production estimates from measurements of spectral radiance and pigment concentrations. *Global Biogeochemical Cycles* 1(3), 171-186, doi:10.1029/GB001i003p00171.
- Campbell, J., Antoine, D., Armstrong, R., Arrigo, K., Balch, W., Barber, R., Behrenfeld, M., Bidigare, R., Bishop, J., Carr, M.-E., Esaias, W., Falkowski, P., Hoepffner, N., Iverson, R., Kiefer, D., Lohrenz, S., Marra, J., Morel, A., Ryan, J., Vedernikov, V., Waters, K., Yentsch, C., Yoder, J., 2002. Comparison of algorithms for estimating ocean primary production from surface chlorophyll, temperature, and irradiance. *Global Biogeochem. Cycles* 16, 9-19-15. <https://doi.org/10.1029/2001GB001444>.
- Croll, D.A., B. Marinovic, S. Benson, F.P. Chavez, N. Black, R. Ternullo, B.R. Tershy, 2005. From wind to whales: trophic links in a coastal upwelling system. *Marine Ecology Progress Series* 289, 117-130.
- Cullen, J.J., 1990. On models of growth and photosynthesis in phytoplankton. *Deep Sea Research Part A: Oceanographic Research Papers* 37(4), 667-683.
- Eppley, R.W., 1972. Temperature and phytoplankton growth in the sea. *Fishery Bulletin* 70(4), 1063-1085.
- Eppley, R.W., E.H. Renger, 1974. Nitrogen assimilation of an oceanic diatom in nitrogen-limited continuous culture. *Journal of Phycology* 10(1), 15-23.
- Eppley, R.W., E. Stewart, M.R. Abbott, U. Heyman, 1985. Estimating ocean primary production from satellite chlorophyll. Introduction to regional differences and statistics for the Southern California Bight. *Journal of Plankton Research* 7(1), 57-70.
- Falkowski, P., 1988. Ocean productivity from space. *Nature* 335(6187), 205-205, doi:10.1039/335205a0.
- Friedrichs, M.A.M., Carr, M.-E., Barber, R.T., Scardi, M., Antoine, D., Armstrong, R.A., Asanuma, I., Behrenfeld, M.J., Buitenhuis, E.T., Chai, F., Christian, J.R., Ciotti, A.M., Doney, S.C., Dowell, M., Dunne, J., Gentili, B., Gregg, W., Hoepffner, N., Ishizaka, J., Kameda, T., Lima, I., Marra, J., Mélin, F., Moore, J.K., Morel, A., O'Malley, R.T., O'Reilly, J., Saba, V.S., Schmeltz, M., Smyth, T.J., Tjiputra, J., Waters, K., Westberry, T.K., Winguth, A., 2009. Assessing the uncertainties of model estimates of primary productivity in the tropical Pacific Ocean. *J. Mar. Syst.* 76, 113-133. <https://doi.org/10.1016/j.jmarsys.2008.05.010>.
- Gall, M.P., I. Hawes, P.W. Boyd, 1999. Predicting rates of primary production in the vicinity of the subtropical convergence east of New Zealand. *New Zealand Journal of Marine and Freshwater Research* 33, 443-455, doi:10.1080/00288330.1999.9516890.

La Fontaine, Y. de, R.H. Peters, 1986. Empirical relationship for marine primary production: the effect of environmental variables. *Oceanologica Acta* 9(1), 65-72.

Morel, A., 1991. Light and marine photosynthesis: A spectral model with geochemical and climatological implications. *Progress in Oceanography* 26(3), 263-306, doi:10.1016/0079-6611(91)90004-6.

Owen, R.W., B. Zeitzschel, 1970. Phytoplankton production: seasonal change in the oceanic eastern tropical Pacific. *Marine Biology* 7, 32-36.

Perry, M.J., 1986. Assessing marine primary production from space. *Bioscience* 36(7), 461-467.

Platt, T., S. Sathyendranath, C.M. Caverhill, M.R. Lewis, 1988. Ocean primary production and available light – further algorithms for remote sensing. *Deep-Sea Research Part A: Oceanographic Research Papers* 35(6), 855-879, doi:10.1016/0198-0149(88)90064-7.

Proud, R., M.J. Cox, A.S. Brierley, 2017. Biogeography of the global ocean's mesopelagic zone. *Current Biology* 27(1),113-119, doi:10.1016/j.cub.2016.11.003.

Ryther, J.H., 1956. Photosynthesis in the ocean as a function of light intensity. *Limnology and Oceanography* 1, 61-70.

Sathyendranath, S., V. Stuart, A. Nair, K. Oka, T. Nakane, H. Bouman, M.-H. Forget, H. Maass, T. Platt, 2009. Carbon-to-chlorophyll ratio and growth rate of phytoplankton in the sea. *Marine Ecology Progress Series* 383, 73-84, doi:10.3354/meps07998.

Steele, J.H., I.E. Baird, 1961. Relations between primary production, chlorophyll and particulate carbon. *Limnology & Oceanography* 6, 68-78.

Steemann Nielsen, E., 1952. The use of radio-active carbon (C14) for measuring organic production in the sea. *Journal du Conseil International pour Exploration de la Mer* 19, 117-140 doi:10.1093/icesjms/18.2.117.

Talling, J.F., 1957. The phytoplankton population as a compound photosynthetic system. *New Phytologist* 56, 133-149.

16. L19-21: *Must reference at least Hanson et al. (2016).*

Response: This reference has been added—line 101 in revised document

17. L31-32: *Stable isotope analysis can, at best, be used to INFER nutrient sources. This limitation is especially pronounced for bulk isotope analyses and un-robust analyses that are not constrained by experimental validation of discrimination factors and isotope baseline variation. The misuse of isotope analyses and reporting of over-interpreted isotope data is a growing problem in the literature and needs to be strongly guarded against.*

Response: This sentence has been adjusted to change the emphasis as suggested—line 108 in revised document.

18. L34: *Why is only one reference published in the last year provided for stable*

isotope studies of reef fish nutrients sources? Please review the literature more widely and reference more appropriately.

Response: Several additional suggested references have been added here—line 110 to 112.

19. L36-38: The text “distinctive basal isotopic signatures dependent on the contribution of differing primary production sources” is very awkward and difficult to understand.

Response: This sentence has been edited to improve clarity—line 110 to 112.

20. L38-43: Referencing needs improving, at least include reference to ideas from, : (Wyatt et al., 2012b; McMahon et al., 2016; Morais & Bellwood, 2019; Skinner et al., 2019).

Response: This section includes Morais & Bellwood 2019 and Wyatt et al. 2012.

21. L53: Please include references for C:N being a good proxy for reef fish condition. Consider robustly defining what is meant by condition.

Response: We provide additional information to specify what is meant by condition and provide reference for C:N as a proxy for reef fish condition—lines 122 to 124 in revised document.

22. L12-23: Why these sites? What is the a-priori information that would lead to a hypothesize of any difference in oceanic nutrient supply between these sites?

Response: We expect these sites to differ in their oceanographic conditions based on their positions encompassing all major cardinal directions around 3 atolls. We expect (as noted by two other reviewers) that there will be differences due to the varying exposure to wind, wave and currents which would affect the delivery of external nutrient subsidies to these reef slope sites. It seems reasonable to expect, based on previous studies (Baines et al., 1994; Behrenfeld & Falkowski, 1997a,b; Croll et al., 2005; Gall et al., 1999; Gove et al. 2016, Proud et al. 2017) that these differences will be translated into chl-a variations across sites.

23. L46: A depth relationship will be next to impossible to robustly determine from sampling at two depths.

Response: We clarify that we are examining for a difference between these two depths, rather than establishing a depth relationship across a range of depths. Our working hypothesis is that there may be differences between the two depths, but this may not be the case—we aim to outline the methods to test this as clearly as possible within the registered report format.

24. L30-34: *There is standard, accepted best practice in reporting isotope analyses that should be adhered to, including terminology (Coplen, 2011; Skrzypek, 2013; Coleman & Meier-Augenstein, 2014). In particular, this text lacks information on the normalization procedure and the identity of standard materials.*

Response: We now report additional information to adhere to these standards of reporting—lines 209 to 211.

25. L20-50: *Bulk tissue stable isotope data CANNOT be interpreted in the absence of detailed quantification of the isotope baselines in space and time and understanding of the role of physiology and metabolism leading to isotopic variation. The proposed analysis has the strong potential to represent another ambiguous coral reef isotope data set with limited robust inference potential.*

Response: As noted in earlier responses, we are including quantification of stable isotope data from POM and zooplankton samples to address this, and to provide information on the spatial variation in the basal isotopic sources. We accept that our ability to characterize temporal variation in isotopic values is limited within the present approach, but hope to add temporal components to future work.

26. L44-51: *Arithmetic lipid normalisation needs better justification, including much wider and up to date referencing (e.g. Skinner et al., 2016). Why apply lipid normalisation rather than lipid removal if there truly is a lipid concern? What was the lipid content of the samples (not the C:N)? What was the verified isotope effect of lipid removal for these samples?*

Response: Mathematical lipid normalization preserves the integrity of samples for $\delta^{15}\text{N}$ analysis (Post 2007). Based on the Skinner et al. (2016) publication, we clarify that arithmetic lipid normalization will only be used if C:N values exceed 3.5.

27. L51-on: *Satellite inferences would seem to need to be ground-truthed to be quantitatively meaningful in this context.*

Response: Whilst the use of satellite-derived chlorophyll-a concentration as a proxy for biomass, productivity or production is associated with uncertainties, the most intransigent being effective photosynthetic efficiency, in situ data are also associated with uncertainties. Logistics alone prevent the measurement of water-column production in situ over the life cycle of a given reef fish; many research vessels (or expensive proxy sensors) would have to be deployed continuously to gather, incubate and measure production rates across each reef. Even the measurement of sufficient in situ samples to validate remote sensing products is challenging (e.g. Campbell et al., 2002; Friedrichs et al., 2009).

In the absence of validation data, we draw on the MODIS-Aqua dataset to validate the Sentinel 3 product. We hope in future to gain funding to use extensive in situ data to validate the OLCI chl product and a representative ensemble of primary production models with which to extrapolate production estimates in space and time, using satellite data, to span the life cycle of reef fish across the Chagos Archipelago (sensu

Bidigare et al., 1987; Falkowski, 1988). In the absence of such resources, there is sufficient support in the literature for the use of satellite chlorophyll as a proxy for primary production (e.g. Baines et al., 1994; Behrenfeld & Falkowski, 1997a,b; Croll et al., 2005; Gall et al., 1999; Proud et al. 2017) provided that uncertainties are considered in the analysis.

2.3. References

- Andrews, J.C. and Gentien, P. (1982) *Upwelling as a source of nutrients to the Great Barrier Reef ecosystem: A solution to Darwin's question? Marine Ecology Progress Series* 8: 257-269.
- Atkinson, M.J. (1992) *Productivity of the Enewetak flats predicted from mass transfer relationships. Continental Shelf Research* 12: 799-807.
- Atkinson, M.J. and Bilger, R.W. (1992) *Effects of water velocity on phosphate uptake in coral reef flat communities. Limnology and Oceanography* 37: 273-279.
- Atkinson, M.J. and Falter, J.L. (2003) *Coral reefs. In: Black, K., Shimmield, G. (eds) Biogeochemistry of marine systems. CRC Press, Boca Raton, Florida, pp 40-64.*
- Coleman, M. and Meier-Augenstein, W. (2014) *Ignoring IUPAC guidelines for measurement and reporting of stable isotope abundance values affects us all. Rapid Communications in Mass Spectrometry* 28: 1953-1955.
- Coplen, T.B. (2011) *Guidelines and recommended terms for expression of stable-isotope-ratio and gas-ratio measurement results. Rapid Communications in Mass Spectrometry* 25: 2538-2560.
- Cyronak, T., Takeshita, Y., Courtney, T.A., DeCarlo, E.H., Eyre, B.D., Kline, D.I., Martz, T., Page, H., Price, N.N., Smith, J., Stoltenberg, L., Tresguerres, M. and Andersson, A.J. (2020) *Diel temperature and pH variability scale with depth across diverse coral reef habitats. Limnology and Oceanography Letters* 5: 193-203.
- Genin, A., Monismith, S.G., Reidenbach, M.A., Yahel, G. and Koseff, J.R. (2009) *Intense benthic grazing of phytoplankton in a coral reef. Limnology and Oceanography* 54: 938-951.
- Grottoli, A.G., Rodrigues, L.J. and Palardy, J.E. (2006) *Heterotrophic plasticity and resilience in bleached corals. Nature* 440: 1186-1189.
- Hamner, W.M., Colin, P.L. and Hamner, P.P. (2007) *Export-import dynamics of zooplankton on a coral reef in Palau. Marine Ecology-Progress Series* 334: 83-92.
- Hamner, W.M., Jones, M.S., Carleton, J.H., Hauri, I.R. and Williams, D.M. (1988) *Zooplankton, planktivorous fish, and water currents on a windward reef face: Great Barrier Reef, Australia. Bulletin of Marine Science* 42: 459-479.
- Hanson, K.M., Schnarr, E.L. and Leichter, J.J. (2016) *Non-random feeding enhances the contribution of oceanic zooplankton to the diet of the planktivorous coral reef fish *Dascyllus flavicaudus*. Marine Biology* 163.
- Leichter, J.J., Helmuth, B. and Fischer, A.M. (2006) *Variation beneath the surface: Quantifying complex thermal environments on coral reefs in the Caribbean, Bahamas and Florida. Journal of Marine Research* 64: 563-588.
- Leichter, J.J., Stokes, M.D., Hench, J.L., Witting, J. and Washburn, L. (2012) *The island-scale internal wave climate of Moorea, French Polynesia. Journal of Geophysical Research-Oceans* 117: C06008.
- Lesser, M.P., Falcon, L.I., Rodriguez-Roman, A., Enriquez, S., Hoegh-Guldberg, O. and Iglesias-Prieto, R. (2007) *Nitrogen fixation by symbiotic cyanobacteria provides a source of nitrogen for the scleractinian coral *Montastraea cavernosa*. Marine*

Ecology-Progress Series 346: 143-152.

McMahon, K.W., Thorrold, S.R., Houghton, L.A. and Berumen, M.L. (2016) Tracing carbon flow through coral reef food webs using a compound-specific stable isotope approach. *Oecologia* 180: 809-821.

Morais, R.A. and Bellwood, D.R. (2019) Pelagic Subsidies Underpin Fish Productivity on a Degraded Coral Reef. *Current Biology* 29: 1521-1527 e1526.

Patten, N.L., Wyatt, A.S.J., Lowe, R.L. and Waite, A.M. (2011) Uptake of picophytoplankton, bacterioplankton and virioplankton by a fringing coral reef community (Ningaloo Reef, Australia). *Coral Reefs* 30: 555-567.

Skinner, C., Newman, S.P., Mill, A.C., Newton, J. and Polunin, N.V.C. (2019) Prevalence of pelagic dependence among coral reef predators across an atoll seascape. *Journal of Animal Ecology* 88: 1564-1574.

Skinner, M.M., Martin, A.A. and Moore, B.C. (2016) Is lipid correction necessary in the stable isotope analysis of fish tissues? *Rapid Communications in Mass Spectrometry* 30: 881-889.

Skrzypek, G. (2013) Normalization procedures and reference material selection in stable HCNOS isotope analyses: an overview. *Analytical and Bioanalytical Chemistry* 405: 2815-2823.

Steele, J.H. and Baird, I.E. (1961) Relations between primary production, chlorophyll and particulate carbon. *Limnology and Oceanography* 6: 68-78.

Wiebe, W.J., Johannes, R.E. and Webb, K.L. (1975) Nitrogen fixation in a coral reef community. *Science* 188: 257-259.

Wyatt, A.S.J., Falter, J.L., Lowe, R.J., Humphries, S. and Waite, A.M. (2012a) Oceanographic forcing of nutrient uptake and release over a fringing coral reef. *Limnology and Oceanography* 57: 401-419.

Wyatt, A.S.J., Leichter, J.J., Toth, L.T., Miyajima, T., Aronson, R.B. and Nagata, T. (2020) Heat accumulation on coral reefs mitigated by internal waves. *Nature Geoscience* 13: 28-34.

Wyatt, A.S.J., Lowe, R.J., Humphries, S. and Waite, A.M. (2010) Particulate nutrient fluxes over a fringing coral reef: relevant scales of phytoplankton production and mechanisms of supply. *Marine Ecology-Progress Series* 405: 113-130.

Wyatt, A.S.J., Lowe, R.J., Humphries, S. and Waite, A.M. (2013) Particulate nutrient fluxes over a fringing coral reef: Source-sink dynamics inferred from carbon to nitrogen ratios and stable isotopes. *Limnology and Oceanography* 58: 409-427.

Wyatt, A.S.J., Waite, A.M. and Humphries, S. (2012b) Stable isotope analysis reveals community-level variation in fish trophodynamics across a fringing coral reef. *Coral Reefs* 31: 1029-1044.

Reviewer: 4

Comments to the Author(s)

The authors propose an investigation of the roles in proximity to deep water nutrients, primary productivity and depth influence the growth and condition of coral reef fishes.

The proposed methods of investigation include; a) fish morphometrics (body length, number of annuali in otoliths), b) elemental chemistry (relative ratios of total and

isotopic weights of C and N) and, c) satellite derived oceanographic data.

Overall, I think it is an interesting and worthy investigation that will contribute toward understanding the effects depth on reef fish ecology - an area of growing interest that has a substantial lack of empirical ecological knowledge.

There are a couple of areas within the proposed methods that I think would make for a clearer and more compelling investigation.

The most important aspects of these are that: 1) I believe there should be greater clarity in the presentation of statistical modelling methods. 2) I have also identified a number of potential confounding factor that could mask positive causal relationships, if not accounted for. Some are likely more critical than others. However, in saying that all but one can be implemented at the data analysis or pre-analysis phases.

Statistical Models:

Pg 9, ln 38: It is generally considered that the variance within random effects is rarely modeled accurately for factors with fewer than 5 - 6 levels. Given that there are only 3 atolls included in this investigation, I would suggest that the authors will have a clearer understanding of their data if they use log likelihood tests to compare models with and without atoll as a fixed factor. Based on the test results, the atoll factor can be formally incorporated in, or dropped from, final models.

In the same context, it would be appropriate and desirable to include site as a random level factor in all models (5-6 total sites at each atoll). If the atoll that fish were collected at significantly affects the data (tested as above), then sites should be nested within atolls.

Response: Whilst we appreciate the value of using log likelihood testing to assess goodness of fit, we feel that carrying out model selection when the data are available, runs against the principle of setting up expectations and models a priori, that the registered report format allows. We therefore feel it is appropriate that we incorporate atoll based on our a priori expectations, and that this remains in the final model, rather than undertaking a model selection process.

Regarding using site as a random effect, we clarify that we cannot include sites as a random effect (simply because we do not have the sample replication at the site level to do this). To illustrate the sampling strategy we have added a sampling diagram (Fig 2), and a table (Table 1) showing numbers of samples obtained.

We appreciate the fact that a RE with less than 5 levels is not ideal, and in response to this we now include atoll as a fixed, rather than a random effect. We clarify that as we are obtaining and analyzing site level mean values for fish isotope and growth metrics and satellite chl-a, we cannot specify site as a random effect.

*Pg 9, ln 46: The construction of the proposed full model is a little unclear in this description. Is there a three-way interaction (meanChl-a*Slope*Depth)? If not, what is the reasoning behind the assumption that an interaction between slope*

and mean Chl-a would not have possible effects on fish growth rate, fish condition , and isotopic value signatures in fish tissues?

Response: We have not specified a three-way interaction, however the suggestion of examining for an interaction between slope and mean chl-a is valuable, and we will now include this in the models. The hypothesis design table has been changed to incorporate this.

Pg 9, ln 52: It seems that using a log likelihood approach comparing the full models proposed in the above paragraph (and atolls as mentioned above) to a depth only model would identify whether or not it is appropriate to pool data from all sites into a depth analysis. I am willing to hear further rationale for taking this pooling approach a-priori.

Response: Based on the this comment, and the comments of reviewer 2 we will not pool data from all sites into a depth analysis, but will constrain the permutation so that blocks of sites are randomly re-assigned, rather than pooling all individual data.

Pg 10, Hypotheses table:

As a note, there must be a clearer way to communicate the main hypotheses. There are 32 listed here.

Response: We have altered the format of the hypotheses table to improve readability.

It looks as though there is sufficient power to address the proposed questions.

I did note, however, that there is not a (clearly labelled) breakdown of the number of samples by depth and site for hypotheses that include depth comparisons. If 7 is the total number of samples across both depths at some sites, the inclusion of these sites may not aid in the constraint of data variance among depths.

Response: A table clearly indicating the number of samples by depth and site is now included (Table 1).

Potential confounding factors that could be tested:

These are some avenues that the investigators might want to explore to better understand their data. This may be particularly important if causal relationships are not found in their primary response variables. It is likely that some of these tests, if adopted, would be best presented in supplemental materials, so as not to detract from the results main investigation.

1) Is there a relationship between variability in the delivery of deep oceanic nutrients and primary productivity? - correlation analyses? formal model interactions, as above?

Response: If we have interpreted this suggestion correctly...we could examine for a relationship between site level variability (standard deviation of OCLI chl-a values) and mean site level chl-a (OCLI chl-a values), and place the results of this within supplemental materials, as suggested. We note we do not have a direct method to

assess variability in the delivery of deep oceanic nutrients, but are relying on satellite chl-a as a proxy for possible external nutrient subsidies.

2) Could nutrient delivery via terrestrial/ lagoonal sources be a confounding factor? The investigators clearly state their sampling was designed to mitigate this effect. I would suggest initially using distance from lagoon entrance/river mouths (there are islands, right?) as a co-variate to check that this potential effect was mitigated. Again, log likelihood tests of models with and without this factor would ascertain any effect.

Response: We appreciate this suggestion, but note several obstacles to carrying out this analysis. Firstly there are no rivers on the islands as a potential confounding source of nutrient delivery. Secondly only one of the atolls has a spatially identifiable lagoon entrance. The other two atolls lack a clearly defined lagoon entrance to measure distance from. We avoided sites at the ends of reef islands to minimize any anomalous effects of localized currents.

3) Exposure: Productivity, surge and wave action (water mixing) all vary with reef exposure. I would suggest formally investigating the potential role of exposure in all models. This could be achieved with three exposure levels (windward, leeward, oblique), or with the cardinal directions if exposure is not known.

Response: We agree that productivity, surge and wave action/mixing are all influenced by reef exposure. We expect that these will be captured within the variability in chl-a values, and we have deliberately located the sites around the cardinal directions of each atoll (as stated in the sampling design).

4) Fish size: fish size can, in some cases, alter C:N ratios (e.g. Sterner and George 2000, Ecology). I suggest regressing FL against C and N metrics to ascertain that this is not a further source of variance in the data for this investigation.

Response: We will carry out this test as suggested, and now state this in the revised text in lines 267 to 270.

5) C:N - lipid ratios: Fagan et al (2011 - Canadian Journal of fish & aquatic science) suggest that multi-taxa, multi-tissue models of C:N - Lipid ratios, such as presented by Post et al. 2007 - and cited in this proposal) may not always hold. Fagan et al. therefore suggest that species level relationships should first be investigated, where possible. If possible to do so, this may give the investigators greater confidence in the conclusions presented by their data.

Response: We agree it would be optimal to investigate a species level C:N to lipid ratio, but it is not possible to do so in this case, given the amount of fish tissue material we were able to obtain.

Further points for clarification:

page 5, ln 46: Please clarify the actual number of samples collected by atoll, site, and depth (via a simple table).

Response: We now provide a table which shows the number of samples collected by atoll, site and depth (Table 1).

Page 8, In 43: What level of spatial accuracy will be achieved in overlaying satellite grids of this resolution on the sites? I am sure the authors have ascertained that the resolution is appropriate for the study, but it would be amiss not to check - given that units are not given for the pixel size.

Response: The spatial resolution of the OLCI imagery from the Sentinel-3A and 3B platforms is 300 m. This is now indicated at this point in the document—line 313 in the revised document.

Journal Name: Royal Society Open Science

Journal Code: RSOS

Online ISSN: 2054-5703

Journal Admin Email: openscience@royalsociety.org

Journal Editor: Andrew Dunn

Journal Editor Email: openscience@royalsociety.org

MS Reference Number: RSOS-201012

Article Status: SUBMITTED

MS Dryad ID: RSOS-201012

MS Title: Linking variation in planktonic primary production to coral reef fish growth and condition

MS Authors: Roche, Ronan; Heenan, Adel; Taylor, Brett; Schwarz, Jill; Fox, Michael; Southworth, Lucy; Williams, Gareth; Turner, John

Contact Author: Ronan Roche

Contact Author Email: r.roche@bangor.ac.uk

Contact Author Address 1:

Contact Author Address 2:

Contact Author Address 3:

Contact Author City:

Contact Author State:

Contact Author Country: United Kingdom of Great Britain and Northern Ireland

Contact Author ZIP/Postal Code:

Keywords: primary production, coral reef fish, Carbon, Nitrogen, Stable isotope analysis, pelagic energetic subsidies

Abstract: Within low nutrient tropical oceans, islands and atolls with higher primary production support higher reef fish biomass and reef organism abundance. External energy subsidies can be delivered onto reefs via a range of physical mechanisms (current-driven upwelling, internal waves, wind-driven mixing). However, the influence of spatial variation in primary production on reef fish growth and condition is largely unknown. In particular, it is not yet clear how variability in food delivery onto a reef interacts with reef depth and slope, and may affect the productivity of reef fish communities. Here we test the hypothesis that with increased proximity to deep-water oceanic allochthonous nutrient sources, or at sites where transportation of these water bodies onto reefs is facilitated by shallower reef slopes, fish growth and condition will be higher, and this pattern will be further emphasised in areas naturally higher in primary production.

Otolith microstructure analysis revealed that reef fish had consistently higher growth at sites with higher mean chlorophyll-a values, and shallower reef slopes. Fish condition, assessed by C:N ratio, increased at sites with higher mean chlorophyll-a values and shallower reef slopes. Fish collected from sites with higher mean chlorophyll-a, and at greater depths across reefs, exhibited enriched $\delta^{15}\text{N}$ and depleted $\delta^{13}\text{C}$ values indicating a reliance on primary production from external deep-water nutrient subsidies. Our results suggest that where deep-water oceanic nutrient sources are delivered onto shallow reefs, they translate into observable increases in individual fish condition and overall population growth.

EndDryadContent

Appendix C

COLEG Y GWYDDORAU NATURIOL
COLLEGE OF NATURAL SCIENCES

YSGOL GWYDDORAU EIGION
SCHOOL OF OCEAN SCIENCES

Registered charity number: 1141565

3rd May 2022

To the Editorial Board of the Royal Society Open Science

Dear Editor,

We thank you for the decision to accept “Linking variation in planktonic primary production to coral reef fish growth and condition” as an in-principle acceptance on 21st May 2021 (Manuscript ID RSOS-201012.R1). We have completed data collection and analysis as approved within this IPA manuscript, and are therefore submitting the Stage 2 manuscript.

We have completed the study as approved using the modified analysis method as agreed in RSOS-201012.R1, and adhered to the registered experimental procedures.

The IPA registered with the Open Science Foundation (<https://osf.io/bgqxx>) is the modified analysis method Report RSOS-201012.R1, the previous IPA had not been registered prior to this modification being approved.

The raw data associated with this study have been archived in the Dryad Digital Repository ([doi:10.5061/dryad.q83bk3jkt](https://doi.org/10.5061/dryad.q83bk3jkt)) and will be made publicly available following Stage 2 review.

Thank you for your time and consideration,

Yours sincerely,

Ronan Roche, Adel Heenan, Brett Taylor, Jill Schwarz, Michael Fox, Lucy Southworth, Gareth Williams, John Turner

PRIFYSGOL BANGOR
YSGOL GWYDDORAU EIGION,
PORTHAETHWY, YNYS MÔN,
LL59 5AB, DU

BANGOR UNIVERSITY
SCHOOL OF OCEAN SCIENCES,
MENAI BRIDGE, ANGLESEY
LL59 5AB, UK

CYMRAD YMCHWIL/RONAN C ROCHE
CANOLFAN GWYDDORAU / CENTRE FOR APPLIED MARINE SCIENCE
RHIF UNIONGYRCHOL / DIRECT LINE: +44 (01248) 383972
EBOST / EMAIL: r.roche@bangor.ac.uk

FFÔN: +44 (01248) 351151

TEL: +44 (01248) 351151

www.bangor.ac.uk

www.bangor.ac.uk/oceansciences